# NEXUS: EXECUTION-GROUNDED MULTI-AGENT TEST ORACLE SYNTHESIS

## ABSTRACT

Test oracle generation in non-regression testing is a longstanding challenge in software engineering, where the goal is to produce oracles that can accurately determine whether a function under test (FUT) behaves as intended for a given input. In this paper, we introduce `Nexus`, a novel multi-agent framework to address this challenge. `Nexus` generates test oracles by leveraging a diverse set of specialized agents that synthesizes test oracles through a structured process of deliberation, validation, and iterative self-refinement. During the deliberation phase, a panel of four specialist agents, each embodying a distinct testing philosophy, collaboratively critiques and refines an initial set of test oracles. Then, in the validation phase, `Nexus` generates a plausible candidate implementation of the FUT and executes the proposed oracles against it in a secure sandbox. For any oracle that fails this execution-based check, `Nexus` activates an automated self-refinement loop, using the specific runtime error to debug and correct the oracle before re-validation. Our extensive evaluation on seven diverse benchmarks demonstrates that `Nexus` consistently and substantially outperforms state-of-the-art baselines. For instance, `Nexus` improves the test-level oracle accuracy on the LiveCodeBench from 46.30% to 57.73% for *GPT-4.1-Mini*. The improved accuracy also significantly enhances downstream tasks: the bug detection rate of *GPT-4.1-Mini* generated test oracles on HumanEval increases from 90.91% to 95.45% for `Nexus` compared to baselines, and the success rate of automated program repair improves from 35.23% to 69.32%.

## 1 INTRODUCTION

Unit testing represents a cornerstone of modern software engineering, enabling developers to verify the correctness of individual code components in isolation (Beck, 2022). A unit test requires two essential components: a **test input** to exercise a specific path within the function under test (FUT), and a **test oracle** to verify the output against expected behavior (Hossain & Dwyer, 2024). However, constructing comprehensive tests remains labor-intensive, demanding deep domain expertise and attention to complex edge cases (Daka et al., 2015; Daka & Fraser, 2014).

To reduce the burden of manual test construction, researchers have developed a wide range of automated test generation techniques, from fuzzing (Fioraldi et al., 2023; Miller et al., 1990), symbolic execution (Cadar et al., 2008; Godefroid et al., 2005), and search-based testing (Fraser & Arcuri, 2011a;b) to modern LLM-based approaches (Du et al., 2024; Qing et al., 2025; Huang et al., 2024). However, these methods largely prioritize generating diverse test inputs for code coverage, while oracle generation remains secondary due to the long-standing *oracle problem* (Barr et al., 2014)—the difficulty of determining correct outputs for non-trivial programs. As a result, most work rely on *regression oracles*, which record outputs from the current implementation as expected behavior. While valuable for detecting behavioral changes, regression oracles fundamentally cannot verify the functional correctness of new or modified code against its intended specification.

The challenge of generating specification-based oracles that verify correctness against intended behavior rather than existing implementation has emerged as a critical bottleneck in automated testing. Moreover, it brings significant obstacles to LLM-based and agent-based automatic code generation, where reliable feedback is essential for enabling LLMs to iteratively refine their outputs.

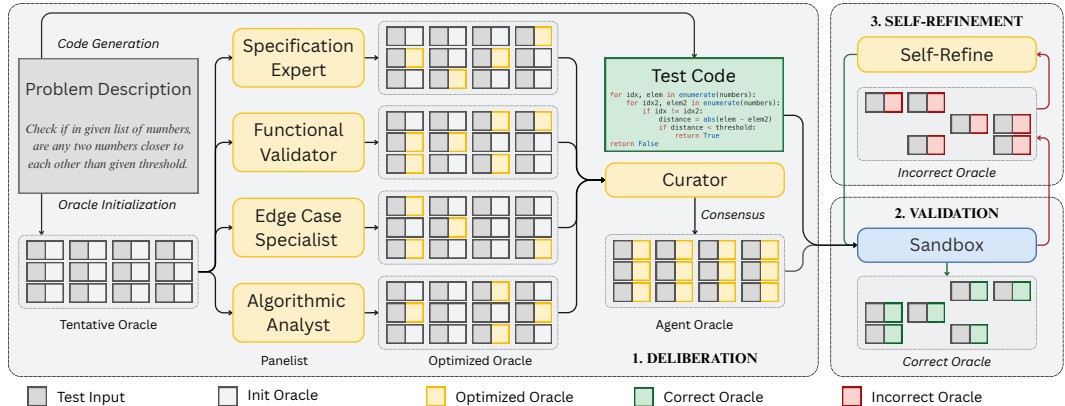

Figure 1: Overview of the `Nexus` framework. **(1) Deliberation:** A panel of specialist agents collaborates to synthesize candidate oracles. **(2) Validation:** These oracles are validated by executing them against LLM-generated plausible implementations of the function under test. **(3) Self-Refinement:** Failed oracles undergo an iterative refinement loop, utilizing runtime errors for automated debugging to ground abstract deliberation in executable evidence.

Recent advances in LLMs have opened new avenues for addressing this challenge, with CANDOR (Xu et al., 2025) representing the state-of-the-art, introducing a multi-agent framework where multiple LLM agents collaborate through panel discussions to reach consensus on oracle predictions. However, our analysis reveals two key limitations of CANDOR: single-paradigm reasoning and a deliberation-only architecture. First, all agents rely on the same source of truth (i.e., the natural language documentation of the FUT), and play similar roles based on identical instructions. This creates an epistemological bottleneck, as the agents share the same interpretation of the specification. Second, reasoning is confined to abstract, language-based deliberation without grounding in execution. Since proposed oracles are never validated against even a plausible implementation, the framework lacks feedback to detect subtle bugs or undocumented edge cases.

We introduce **Nexus**, a multi-agent framework that grounds language-based reasoning in executable evidence. First, to break the analysis echo chamber (Cinelli et al., 2021), the framework employs a deliberation phase with four specialist agents: a **Specification Expert** for literal compliance, an **Edge Case Specialist** for adversarial robustness, a **Functional Validator** for high-level semantic properties, and an **Algorithmic Analyst** for implementation-level logic. This orthogonal approach ensures a far more comprehensive critique than standard re-examination. Second, `Nexus` introduces a validation phase by generating a candidate implementation and executing proposed oracles in a sandbox. Finally, a self-refinement loop utilizes runtime errors to correct failed oracles.

Extensive evaluation on seven benchmarks demonstrates that `Nexus` substantially outperforms state-of-the-art baselines across various open-source and proprietary models. Notably, when powered by *GPT-4.1-Mini*, `Nexus` boosts test-level oracle accuracy on LiveCodeBench to 57.73% from 46.30% achieved by CANDOR. This superiority extends to downstream tasks: on HumanEval, `Nexus` raises bug detection rates from 90.91% to 95.45% and automated program repair success from 35.23% to 69.32%. By combining multi-agent deliberation with validation and self-refinement, `Nexus` establishes a new paradigm for generating reliable, specification-grounded test oracles. The main contributions of this work are as follows:

- We design and implement `Nexus`, a novel multi-agent framework that synthesizes test oracles through a unique pipeline of deliberation, validation, and self-refinement, overcoming the single-source-of-truth limitation in prior work.

- We introduce a panel of four specialist agents, each embodying a distinct testing philosophy, to facilitate a more comprehensive and effective deliberation process for oracle generation.

- We conduct a comprehensive evaluation on seven benchmarks using four different LLMs, demonstrating that `Nexus` significantly outperforms state-of-the-art methods in oracle accuracy and enhances downstream tasks like bug detection and automated program repair.

## 2 BACKGROUND AND RELATED WORK

### 2.1 BACKGROUND

**Unit Testing and the Oracle Problem.** Unit testing serves as a foundational practice in software engineering, designed to validate that individual software components behave as intended under various conditions (Myers et al., 2011; Runeson, 2006). A complete unit test consists of two distinct but equally critical components: the *test input*, which drives the execution of the FUT, and the *test oracle*, which determines whether the execution result matches the expected behavior (Barr et al., 2014). While the field of Automated Test Generation (ATG) has achieved significant maturity in synthesizing test inputs, the automated generation of test oracles remains a formidable challenge, widely referred to as the *Oracle Problem* (Barr et al., 2014). Without a reliable oracle, even high-coverage test inputs are limited to detecting only implicit failures (e.g., crashes or uncaught exceptions), failing to identify functional logic errors where the program executes successfully but produces incorrect results.

**Regression vs. Specification-Based Oracles.** To mitigate the oracle problem, traditional ATG tools predominantly rely on *regression oracles*, which capture the current behavior of the code as the ground truth (Fraser & Arcuri, 2011b; Pacheco & Ernst, 2007). While effective for regression testing, i.e., ensuring that code changes do not alter existing behavior, regression oracles are fundamentally incapable of verifying the correctness of new or buggy code against its intended specification. Consequently, the generation of *specification-based oracles*, which verify correctness against intended behavior (e.g., natural language documentation) rather than the implementation itself, has emerged as a critical research frontier. Prior to the advent of LLMs, generating such oracles required formal specifications, which are rarely available in practice (Daka & Fraser, 2014).

**LLM-Driven Oracle Generation.** Recent advancements in LLMs have spurred a new wave of research aiming to address the specification-based oracle problem by leveraging the model's ability to understand natural language requirements. Early works such as TOGA (Dinella et al., 2022) demonstrated that pre-trained models could infer assertions from code context and docstrings. Subsequent approaches have attempted to improve accuracy through hybrid techniques; for instance, TOGLL (Hossain & Dwyer, 2024) combines EvoSuite-generated inputs with LLM-generated oracles. Most recently, frameworks like CANDOR (Xu et al., 2025) have introduced multi-agent collaboration to reduce hallucinations in oracle prediction. Despite these efforts, generating precise oracles for complex, real-world functions remains an open challenge, as models frequently struggle to align the generated assertions with the subtle logic of the implementation, necessitating more robust verification mechanisms such as the execution-grounded approach proposed in this work.

### 2.2 RELATED WORK

The generation of test oracles presents fundamentally different challenges from input generation, requiring a deep understanding of intended program behavior rather than mere code coverage. Early automated testing tools largely sidestepped this challenge by employing regression oracles or simple crash oracles (Fraser & Arcuri, 2011b; Pacheco & Ernst, 2007). While regression oracles effectively detect behavioral changes between versions, they cannot validate functional correctness against specifications, potentially perpetuating bugs across software evolution (Pizzorno & Berger, 2024). The specification oracle problem that generating oracles that verify correctness against intended behavior has long been recognized as one of the most challenging aspects of automated testing (Barr et al., 2014). The emergence of LLM-based approaches brought new possibilities for oracle generation. TOGA (Dinella et al., 2022) pioneered the use of transformer models for inferring specification-based oracles, fine-tuning CodeBERT to predict expected outputs from code context. This work inspired subsequent research exploring various neural architectures and training strategies for oracle generation (Endres et al., 2024; Hayet et al., 2024). TOGLL (Hossain & Dwyer, 2024) achieved state-of-the-art performance by combining EvoSuite-generated inputs with fine-tuned LLM oracles, demonstrating that hybrid approaches could leverage the strengths of both traditional and learning-based methods. However, fine-tuning approaches face significant limitations in generalization, as model performance degrades substantially when test distributions shift from training data (Xu et al., 2025).

Table 1: Error Distribution (%) of the Incorrect Test Cases across Benchmarks.

| Error | HumanEval | MBPP | MBPP-Pro | HumanEval-Pro | TestEval | LiveCodeBench | ULT |
|---|---|---|---|---|---|---|---|
| Input | 11.11 | 15.38 | 17.14 | 8.57 | 0.00 | 20.29 | 9.38 |
| Output | 88.89 | 84.62 | 82.86 | 91.43 | 100.00 | 79.71 | 90.62 |

Recent work has increasingly focused on prompt engineering that leverage the zero-shot capabilities of LLMs. For instance, Siddiq et al. (2024) demonstrated that carefully crafted prompts enable LLMs to generate both inputs and oracles from natural language descriptions without fine-tuning. RetriGen (Zhang et al., 2025b) enhanced oracle generation through retrieval-augmented generation, incorporating relevant assertions from external codebases to guide prediction. Currently, CANDOR (Xu et al., 2025) represents the state-of-the-art, which introduces a multi-agent framework where several LLM agents collaborate through a panel discussion to reach a consensus on the correct oracle. While effective without fine-tuning, CANDOR's deliberation-only approach struggles to detect bugs beyond explicit specifications.

In this work, we propose `Nexus`, which overcomes these limitations by introducing our novel deliberation-validation-refinement architecture. `Nexus` employs a multi-agent framework that facilitates diverse reasoning strategies, enhancing the initial analysis of test oracles. By incorporating a validation phase that grounds oracles in executable contexts and a self-refinement loop that leverages runtime feedback, `Nexus` further improves the accuracy of generated oracles.

## 3 METHODOLOGY

### 3.1 PRELIMINARY ANALYSIS

To empirically validate our hypothesis that **oracle generation constitutes the primary bottleneck in automated test generation with LLM**s, we conducted a preliminary case study. Specifically, we prompted *GPT-4.1-nano* to generate complete unit test cases from the natural language descriptions across diverse benchmarks. Subsequently, we performed a manual root cause analysis for every incorrectly generated test case, and categorized errors as stemming from either *invalid inputs* or *incorrect oracles.* The evaluation results are shown in Tab. 1, demonstrating a clear and consistent trend across all seven benchmarks, i.e., the majority of errors are attributable to incorrect test oracles. For instance, on the HumanEval benchmark, 88.89% of errors were due to an incorrect oracle, compared to only 11.11% from invalid test inputs. This finding strongly suggests that **while LLMs can proficiently generate syntactically valid and plausible inputs, they struggle profoundly with the deep semantic reasoning required to predict the correct output.**

### 3.2 FRAMEWORK OVERVIEW

To address this critical oracle problem, we introduce `Nexus`, a multi-agent framework that synthesizes test oracles through a structured process of deliberation, validation, and iterative self-refinement. As shown in Fig. 1, unlike prior works that rely on a single source of truth, `Nexus` orchestrates a comprehensive pipeline where candidate oracles are first debated by a panel of specialist agents, then consolidated by a curator, and finally subjected to rigorous execution-based testing. The core novelty of our framework lies in its final two stages: a validation phase that checks oracle correctness against a plausible, LLM-generated implementation of the FUT, and a self-refinement loop that uses execution feedback to automatically debug and correct any oracles that fail this check. This architecture grounds the abstract reasoning of LLMs in concrete, executable evidence, creating an effective feedback mechanism that significantly enhances test oracle accuracy.

### 3.3 MULTI-PHASE ARCHITECTURE

The `Nexus` framework operates as a structured, three-phase pipeline: multi-agent deliberation, execution-based validation, and iterative self-refinement. Each phase progressively scrutinizes and improves a set of candidate oracles, culminating in a final set of validated and self-corrected test oracles. We detail each phase of the pipeline below:

### 3.3.1 PHASE 1: MULTI-AGENT DELIBERATION.

The process begins by generating a set of *Tentative Oracles* using a standard zero-shot prompting approach. These initial oracles serve as concrete hypotheses for a structured, hierarchical debate among a panel of four specialist agents, each embodying a specific testing persona to ensure diverse and comprehensive analysis.

The **Specification Expert** focuses on adherence to documented specifications and requirements, ensuring strict fidelity to the function's explicit contract. It systematically cross-references each candidate oracle against the formal requirements extracted in Phase 1, verifying adherence to documented constraints such as return types, value ranges, and precisely defined behaviors.

The **Edge Case Specialist** focuses on boundary conditions, corner cases, and error scenarios. It probes the oracle's validity under boundary conditions and with atypical inputs often overlooked in documentation, such as empty collections, null values, zeros, and data type extremities.

The **Functional Validator** focuses on core functionality and expected input-output relationships. It concentrates on the core purpose of the function, ensuring the input-output relationship is logically sound and aligns with general programming principles, even for behaviors not explicitly detailed. For instance, for a sort function, it would verify that the output is a permutation of the input, a fundamental property that may not be explicitly stated in the documentation.

The **Algorithmic Analyst** predicts the correct output by performing a conceptual execution of a plausible algorithm. Based on the function's description, it infers a likely implementation strategy. It performs a step-by-step mental trace for a given test input to compute the expected result, making it effective at catching subtle off-by-one or logic errors.

Finally, a *Curator* agent then synthesizes these multiple perspectives, resolves disagreements, and produces a single, consolidated set of *Candidate Oracles* that represents the collective judgment of the deliberation process.

### 3.3.2 PHASE 2: EXECUTION-BASED VALIDATION

To ground the abstract deliberation in concrete evidence, this phase tests the candidate oracles against a plausible implementation. First, an LLM is prompted to generate a *Candidate Implementation* of the FUT based on its documentation. This generated code serves as a reasonable, albeit not guaranteed to be perfect, model of the intended behavior. The candidate oracles are then executed as test assertions against this implementation in a secure *Code Execution Sandbox*. This step provides empirical feedback: oracles that pass are considered highly likely to be correct, while those that fail are flagged as erroneous and passed to the next phase.

### 3.3.3 PHASE 3: ITERATIVE SELF-REFINEMENT

For every oracle that fails validation, an automated refinement loop is activated. The framework constructs a detailed debugging prompt containing the FUT specification, the candidate implementation, the failed oracle, and the specific error message from the sandbox. An LLM is then prompted to generate a corrected oracle. This refinement process is iterative: the proposed fix is sent back to the *Code Execution Sandbox* for re-validation. This loop continues for a predefined number of iterations or until the oracle passes, ensuring the final oracles have successfully withstood rigorous, execution-based scrutiny.

## 4 EVALUATION

Our evaluation is structured to address three key research questions:

**RQ1** investigates how `Nexus` performs in terms of oracle accuracy compared to state-of-the-art baselines across different models and benchmarks.

**RQ2** analyzes the contributions of the core components of the `Nexus` framework to its overall performance.

**RQ3** examines how the number of iterative refinement steps in the self-correction loop impacts the final oracle accuracy.

## 4.1 EXPERIMENTAL SETUP

**Task Definition**   We focus on the task of test oracle generation. In this task, the model is provided with the natural language description of a function (e.g., its docstring) and a predefined test input. The goal is for the model to generate the corresponding test oracle, which is the expected output for that given input, typically formatted as a programmatic assertion. To simulate a fully automated testing pipeline, we use test inputs that are themselves generated by *GPT-4.1-nano*[1].

**Datasets** To ensure our findings are generalizable, we conduct experiments across seven diverse and widely-used benchmarks for code intelligence. HumanEval (Chen et al., 2021) and MBPP (Austin et al., 2021) are canonical benchmarks for evaluating code generation, containing a range of programming problems. We also include their more challenging counterparts, HumanEval-Pro and MBPP-Pro (Yu et al., 2024), which feature more complex requirements and edge cases. Furthermore, we incorporate LiveCodeBench (Jain et al., 2024b), a benchmark designed to be robust against data contamination from web-scale training sets. Finally, to specifically assess performance on testing-related tasks, we include TestEval (Wang et al., 2025) and UnLeakedTestbench (ULT) (Huang et al., 2025), which are benchmarks created explicitly for evaluating test generation capabilities.

**Models** To demonstrate the broad applicability of our framework, we evaluate its performance on a suite of both open-source and closed-source LLMs, representing a range of sizes and capabilities. Our selection of open-source models includes DeepSeek-R1-Distill-Qwen-7B (**Qwen-7B**) and DeepSeek-R1-Distill-Qwen-32B (**Qwen-32B**). For closed-source models, we use OpenAI's **GPT-4.1-nano** and **GPT-4.1-mini**. For all experiments, we set the decoding temperature parameter to $0.0$ to ensure deterministic and reproducible outputs.

**Baselines** To contextualize Nexus's performance, we compare it against two primary baselines. First, **Direct Generation** represents the standard zero-shot prompting approach, where the LLM generates the test oracle directly without any advanced multi-agent framework. This serves as a proxy for naive test generation practices. Second, we compare against **CANDOR** (Xu et al., 2025), the current state-of-the-art multi-agent framework for test oracle generation. As detailed earlier, CANDOR employs a consensus-based panel discussion where agents reason from a single specification source, representing a strong contemporary baseline.

**Evaluation Metrics** Our primary metric is **Oracle Accuracy**, which measures the percentage of generated test oracles that are semantically correct, as determined by execution against the canonical benchmark solution. We report this metric at two levels of granularity. *Task-level Accuracy* applies a strict criterion where a task is considered successful only if *all* generated oracles are correct; otherwise, the task is marked as a failure. Conversely, *Test-level Accuracy* evaluates each oracle individually, calculating the proportion of correct test cases relative to the total number generated per task. To assess practical utility, we also measure the **Bug Detection Rate**, defined as the percentage of buggy programs from a curated set that are successfully identified by our generated test cases. Finally, we evaluate the downstream impact on automated program repair with **Self-Debugging Performance**, measured by the final Pass@1 rate of repaired code against the official hidden test suite after receiving feedback from a generated oracle.

## 5 RESULTS

### 5.1 RQ1: ORACLE GENERATION ACCURACY

To answer our first research question, we compared the oracle accuracy of Nexus against both Direct Generation and the CANDOR framework across all selected models and benchmarks. The evaluation results are shown in Tab. 2, measured at both the task level (whether all oracles for a task are correct) and the more granular test-case level. The evaluation results demonstrate that Nexus consistently and substantially outperforms both baselines. For instance, when using the powerful *GPT-4.1-Mini* model, Nexus achieves a test-level accuracy of 93.69% on HumanEval. This represents a significant improvement of 13.45 absolute points over the 80.24% from Direct Generation and also surpasses CANDOR's 91.95%. This trend of superior performance holds across nearly

---

[1]For each programming task, we generate 20 diverse test inputs and keep them fixed across all experiments to ensure fair and consistent comparisons. These inputs are available in our replication package.

Table 2: Accuracy (%) of LLM-generated test oracles at both the *Task-Level* and *Test-Level*. The value in parentheses denotes the absolute improvement via Direct Generation. **HE**: HumanEval, **HE-Pro**: HumanEval-Pro, **TE**: TestEval, **LCB**: LiveCodeBench, **ULT**: UnLeakedTestbench.

| Method | HE | HE-Pro | MBPP | MBPP-Pro | TE | LCB | ULT |
|---|---|---|---|---|---|---|---|
| **Task-Level Accuracy (%)** | | | | | | | |
| Qwen-7B | 10.98 | 1.22 | 14.40 | 3.70 | 0.00 | 0.93 | 0.00 |
| + CANDOR | 6.10 (-4.88) | 1.83 (+0.61) | 14.79 (+0.39) | 2.65 (-1.06) | 0.49 (+0.49) | 0.00 (-0.93) | 0.00 (+0.00) |
| + Nexus | **13.41 (+2.43)** | **3.66 (+2.44)** | **20.23 (+5.83)** | **5.03 (+1.33)** | **0.49 (+0.49)** | **3.74 (+2.81)** | **0.85 (+0.85)** |
| Qwen-32B | 12.80 | 4.27 | 24.51 | 9.79 | 0.00 | 0.93 | 1.71 |
| + CANDOR | 6.10 (-6.70) | 3.66 (-0.61) | 7.39 (-17.12) | 2.65 (-7.14) | 0.00 (+0.00) | 8.41 (+7.48) | 0.85 (-0.86) |
| + Nexus | **29.88 (+17.08)** | **10.37 (+6.10)** | **30.74 (+6.23)** | **14.81 (+5.02)** | **0.00 (+0.00)** | **9.35 (+8.42)** | **1.71 (+0.00)** |
| GPT-4.1-Nano | 16.46 | 4.88 | 28.02 | 11.38 | 1.95 | 1.87 | 0.00 |
| + CANDOR | 23.78 (+7.32) | 4.27 (-0.61) | 29.57 (+1.55) | 12.17 (+0.79) | 1.46 (-0.49) | 1.87 (+0.00) | 0.00 (+0.00) |
| + Nexus | **28.66 (+12.20)** | **8.54 (+3.66)** | **36.58 (+8.56)** | **18.25 (+6.87)** | **6.83 (+4.88)** | **1.87 (+0.00)** | **1.71 (+1.71)** |
| GPT-4.1-Mini | 27.44 | 10.37 | 37.35 | 19.05 | 3.41 | 4.67 | 0.85 |
| + CANDOR | 56.10 (+28.66) | 28.66 (+18.29) | 45.91 (+8.56) | 34.92 (+15.87) | 11.22 (+7.81) | 3.74 (-0.93) | 0.85 (+0.00) |
| + Nexus | **65.85 (+38.41)** | **41.46 (+27.44)** | **56.42 (+16.73)** | **42.59 (+21.16)** | **32.20 (+26.34)** | **16.82 (+12.15)** | **5.98 (+5.13)** |
| **Test-Level Accuracy (%)** | | | | | | | |
| Qwen-7B | 27.65 | 10.62 | 31.51 | 13.95 | 1.36 | 10.43 | 8.59 |
| + CANDOR | 26.55 (-1.10) | 12.88 (+2.26) | 30.66 (-0.85) | 12.47 (-1.48) | 2.77 (+1.41) | 10.43 (+0.00) | 8.93 (+0.34) |
| + Nexus | **62.56 (+34.91)** | **41.98 (+31.36)** | **60.81 (+29.30)** | **47.16 (+33.21)** | **16.93 (+15.57)** | **17.07 (+6.64)** | **10.09 (+1.50)** |
| Qwen-32B | 45.64 | 22.01 | 53.03 | 34.86 | 1.81 | 9.69 | 9.24 |
| + CANDOR | 18.75 (-26.89) | 14.18 (-7.83) | 17.02 (-36.01) | 10.87 (-23.99) | 7.92 (+6.11) | 35.51 (+25.82) | 11.11 (+1.87) |
| + Nexus | **76.62 (+30.98)** | **56.97 (+34.96)** | **73.57 (+20.54)** | **61.37 (+26.51)** | **25.01 (+23.20)** | **37.20 (+27.51)** | **13.93 (+4.69)** |
| GPT-4.1-Nano | 72.35 | 52.51 | 69.32 | 54.49 | 41.54 | 31.11 | 14.02 |
| + CANDOR | 79.05 (+6.70) | 55.26 (+2.75) | 72.05 (+2.73) | 59.27 (+4.78) | 49.70 (+8.16) | 32.58 (+1.47) | 13.29 (-0.73) |
| + Nexus | **80.79 (+8.44)** | **57.55 (+5.04)** | **75.24 (+5.92)** | **63.03 (+8.54)** | **57.06 (+15.52)** | **35.09 (+3.98)** | **17.82 (+3.80)** |
| GPT-4.1-Mini | 79.85 | 63.10 | 78.05 | 65.70 | 55.35 | 45.52 | 19.53 |
| + CANDOR | 91.95 (+12.10) | 75.29 (+12.19) | 83.73 (+5.68) | 75.53 (+9.83) | 66.47 (+11.12) | 46.30 (+0.78) | 20.85 (+1.32) |
| + Nexus | **93.69 (+13.45)** | **79.47 (+18.02)** | **86.64 (+8.55)** | **77.40 (+12.38)** | **80.92 (+25.76)** | **56.71 (+11.14)** | **23.21 (+3.68)** |

all settings, indicating that our framework generates more accurate oracles regardless of the underlying LLM or the target benchmark. Notably, the performance gap is particularly pronounced for more capable models. With Qwen-32B on the MBPP benchmark, Nexus improves the test-level accuracy from a baseline of 53.03% to 73.57%, an increase of over 20 absolute points. In the same setting, CANDOR's performance degrades significantly, suggesting that Nexus's structured pipeline is more effective at harnessing the advanced reasoning capabilities of powerful LLMs. Furthermore, on challenging benchmarks designed specifically for testing, such as TestEval and ULT, Nexus often turns near-zero accuracy from baselines into meaningful, double-digit performance gains, establishing a new state-of-the-art in these domains.

## 5.2 RQ2: Ablation Study on Framework Components

Next, to understand the sources of Nexus's performance gains and answer RQ2, we conducted two ablation studies. First, we analyzed the high-level contributions of the multi-agent deliberation (Planning) and execution-based self-correction (Refinement) phases. Second, we performed a more granular analysis to evaluate the individual importance of each agent within the deliberation phase. The results of the phase-level ablation are presented in Tab. 3. We observe that each component plays a critical and complementary role in achieving the final performance. Adding only the self-refinement phase (+ Refinement) to the baseline provides a consistent boost, improving test-level accuracy on HumanEval from 80.24% to 85.24%. However, adding only the multi-agent deliberation phase (+ Planning) yields an even greater improvement, raising the accuracy to 91.68%. This highlights the profound impact of the structured debate among specialized agents in generating high-quality initial hypotheses. The full Nexus framework, which combines both deliberation and refinement, consistently achieves the highest accuracy, demonstrating that its performance stems from the synergistic combination of its deliberative reasoning and validation stages.

To further dissect the deliberation phase, we evaluated variants of Nexus where each of the four Panelist agents was individually removed. Our agent selection was informed by software engineering best practices, identifying four distinct perspectives commonly used to derive test oracles: specification compliance, edge case analysis, functional correctness, and algorithmic verification. As shown in Tab. 4, the evaluation results reveal that every agent contributes positively to the outcome, as removing any one of them degrades performance. Notably, the removal of the **Algorithmic**

Table 3: Phase-level ablation study on the components of `Nexus` using the *GPT-4.1-Mini* model. We report test-level accuracy (%).

| Method | HE | HE-Pro | MBPP | MBPP-Pro | TE | LCB | ULT |
|---|---|---|---|---|---|---|---|
| GPT4.1-Mini | 80.24 | 61.46 | 78.09 | 65.02 | 55.15 | 45.57 | 19.53 |
| + Refinement Only | 85.24 (+5.00) | 68.51 (+7.06) | 82.08 (+3.99) | 69.77 (+4.75) | 73.39 (+18.23) | 54.05 (+8.48) | 21.24 (+1.71) |
| + Planning Only | 91.68 (+11.43) | 76.01 (+14.55) | 85.12 (+7.02) | 75.39 (+10.37) | 59.05 (+3.89) | 45.38 (-0.20) | 21.75 (+2.22) |
| Nexus (Full) | **93.69 (+13.45)** | **79.47 (+18.02)** | **86.64 (+8.55)** | **77.40 (+12.38)** | **80.92 (+25.76)** | **56.71 (+11.14)** | **23.21 (+3.68)** |

Table 4: Agent-level ablation study on the Panelists within `Nexus` using *GPT-4.1-Mini*. Performance is reported at the test-case level (%). **SE**: Specification Expert, **ES**: Edge Case Specialist, **FV**: Functional Validator, **AA**: Algorithmic Analyst.

| Method | HE | HE-Pro | MBPP | MBPP-Pro | TE | LCB | ULT |
|---|---|---|---|---|---|---|---|
| Nexus | **93.69** | **79.47** | **86.16** | **76.88** | **80.92** | **56.71** | **23.12** |
| w/o SE | 91.62 (-2.07) | 73.47 (-6.01) | 84.66 (-1.50) | 75.55 (-1.33) | 79.29 (-1.62) | 54.88 (-1.83) | 21.96 (-1.16) |
| w/o ES | 91.22 (-2.47) | 74.18 (-5.29) | 84.26 (-1.89) | 75.23 (-1.65) | 80.46 (-0.46) | 55.79 (-0.92) | 21.62 (-1.50) |
| w/o FV | 90.88 (-2.80) | 74.24 (-5.23) | 84.96 (-1.20) | 73.54 (-3.34) | 76.47 (-4.45) | 55.90 (-0.81) | 21.27 (-1.85) |
| w/o AA | 89.36 (-4.33) | 72.45 (-7.03) | 83.86 (-2.30) | 72.59 (-4.29) | 77.53 (-3.39) | 55.71 (-1.00) | 21.50 (-1.62) |

**Analyst** (w/o AA) and the **Functional Validator** (w/o FV) results in the most significant drops in accuracy across most benchmarks. For example, on HumanEval-Pro, removing the Algorithmic Analyst reduces test-level accuracy from 79.47% to 72.45%, a drop of over 7 points. This finding underscores the importance of incorporating diverse, orthogonal reasoning strategies. While the Specification and Edge Case agents are crucial for correctness, the functional and algorithmic perspectives appear to be most effective at identifying and correcting subtle logical flaws in the initial oracles, confirming that the strength of the deliberation phase lies in its diversity.

To assess the necessity of role specialization, we investigated whether generic agents could achieve comparable performance. We conducted an ablation study on HumanEval and HumanEval-Pro, replacing each specialized agent with a generic "common panelist" prompted only to generate assertions. As shown in Tab. 9, while a generic replacement outperforms removing a component entirely—for instance, replacing the Algorithmic Analyst improves HumanEval accuracy from 89.36% to 92.74%—the full `Nexus` framework with specialized roles still achieves the highest performance (93.69%). This confirms that the specific, orthogonal testing philosophies embodied by our agents provide a significant advantage over generic reasoning.

### 5.3 RQ3: IMPACT OF REFINEMENT ITERATIONS

The self-refinement phase in `Nexus` is iterative, allowing the framework to make multiple attempts at correcting a failed oracle. To investigate the impact of this iterative process and answer RQ3, we varied the number of refinement loops from 0 (no refinement) to 5. Tab. 5 shows the test-level accuracy for *GPT-4.1-Mini* as a function of the number of refinement steps. We can observe that the evaluation results demonstrate a positive correlation between the number of refinement steps and oracle accuracy, though with diminishing returns. The most substantial performance gains typically occur in the first iteration. For example, on the TestEval benchmark, accuracy jumps from 59.05% with 0 iterations to 78.66% with just one, demonstrating the immediate value of a single round of execution-based feedback. Performance continues to climb with additional steps, generally beginning to plateau after 3 to 5 iterations. On HumanEval-Pro, for instance, accuracy improves from 76.01% (0 iterations) to 79.47% (5 iterations). Based on these findings, we use 5 refinement steps as the default configuration in all other experiments to maximize oracle accuracy.

## 6 DISCUSSION

### 6.1 EFFECTIVENESS IN BUG DETECTION

A primary application of a test oracle is to serve as a ground truth for identifying incorrect program behaviors, or bugs. To measure this practical utility, we evaluated the bug detection capability of the oracles generated by `Nexus` against our curated *Buggy Code Set*. The evaluation results are shown in Tab. 6, which demonstrate that the higher accuracy of `Nexus`-generated oracles directly translates into a higher bug detection rate compared to baseline approaches. For instance, when us-

Table 5: Impact of the number of self-refinement iterations on test-case level oracle accuracy for GPT-4.1-Mini. The 0 iteration row represents Nexus's performance after the deliberation phase and initial validation, before any corrective refinement loops are run. Deltas are shown relative to the Direct Generation baseline.

| Iterations | HE | HE-Pro | MBPP | MBPP-Pro | TE | LCB | ULT |
|---|---|---|---|---|---|---|---|
| Direct | 80.24 | 61.46 | 78.09 | 65.02 | 55.15 | 45.57 | 19.53 |
| Nexus (0) | 91.68 (+11.43) | 76.01 (+14.55) | 85.12 (+7.02) | 75.39 (+10.37) | 59.05 (+3.89) | 45.38 (-0.20) | 21.75 (+2.22) |
| Nexus (1) | 93.26 (+13.02) | 77.24 (+15.79) | 85.77 (+7.68) | 76.99 (+11.96) | 78.66 (+23.51) | 54.79 (+9.21) | 22.82 (+3.29) |
| Nexus (3) | 93.05 (+12.80) | 78.27 (+16.81) | 86.16 (+8.06) | 76.88 (+11.86) | 79.08 (+23.93) | 56.53 (+10.95) | 23.12 (+3.59) |
| Nexus (5) | **93.69 (+13.45)** | **79.47 (+18.02)** | **86.64 (+8.55)** | **77.40 (+12.38)** | **80.92 (+25.76)** | **56.71 (+11.14)** | **23.21 (+3.68)** |

Table 6: Bug Detection Rate (%). We report the percentage of buggy programs successfully identified by test cases using the generated oracles (for correct oracles only). Deltas are shown relative to the Direct Generation baseline for each model.

| Method | HE | HE-Pro | MBPP | MBPP-Pro | TE | LCB | ULT |
|---|---|---|---|---|---|---|---|
| **Underlying Model: GPT-4.1-Nano** | | | | | | | |
| Direct Generation | 43.18 | 48.39 | 50.16 | 45.78 | 22.22 | 6.64 | 36.61 |
| CANDOR | 55.68 (+12.50) | 49.19 (+0.80) | 57.73 (+7.57) | 60.88 (+15.10) | 63.70 (+41.48) | 13.29 (+6.65) | 33.33 (-3.28) |
| Nexus | **81.82 (+38.64)** | **59.27 (+10.88)** | **72.24 (+22.08)** | **71.46 (+25.68)** | **80.00 (+57.78)** | **26.58 (+19.94)** | **37.16 (+0.55)** |
| **Underlying Model: GPT-4.1-Mini** | | | | | | | |
| Direct Generation | 60.23 | 60.89 | 61.83 | 49.70 | 35.93 | 34.55 | 39.34 |
| CANDOR | 90.91 (+30.68) | 64.52 (+3.63) | 64.35 (+2.52) | 64.33 (+14.63) | 81.11 (+45.18) | 35.55 (+1.00) | 37.16 (-2.18) |
| Nexus | **95.45 (+35.22)** | **64.92 (+4.03)** | **76.66 (+14.83)** | **74.55 (+24.85)** | **81.85 (+45.92)** | **42.52 (+7.97)** | **45.90 (+6.56)** |

ing *GPT-4.1-Mini*, test cases equipped with Nexus oracles successfully identified $95.45\%$ of buggy programs for HumanEval tasks. This represents a substantial improvement over the $60.23\%$ detection rate achieved with Direct Generation and also surpasses CANDOR's $90.91\%$. We attribute this improved performance to Nexus's multi-paradigm approach. By integrating specification-based reasoning, edge case analysis, functional validation, and algorithmic simulation, Nexus generates correct oracles for complex scenarios where bugs frequently appear. These are precisely the cases that are often missed by methods that rely solely on documentation, which may be incomplete or lack explicit descriptions of such edge conditions.

## 6.2 DOWNSTREAM IMPACT ON SELF-DEBUGGING

Beyond merely identifying bugs, high-quality test oracles can play a crucial role in guiding automated program repair. We investigated this downstream impact by simulating a single-turn self-debugging loop, where an LLM attempts to fix a buggy program after receiving feedback from a failing test case. The ultimate success is measured by the Pass@1 rate of the repaired code against the benchmark's hidden test suite, with results presented in Tab. 7. We clarify that this evaluation is conducted on our constructed buggy code set, where each solution is verified to be incorrect. Therefore, the baseline performance without debugging information is 0% for all datasets. The findings reveal the significant downstream value provided by Nexus. Using *GPT-4.1-Mini*, the Pass@1 rate on HumanEval after one debugging round with Nexus-generated feedback reached $69.32\%$. This is a dramatic leap from the $28.41\%$ baseline achieved with oracles from Direct Generation. The high-fidelity feedback provided by Nexus's accurate test cases furnishes the LLM with a clear and unambiguous error signal, enabling it to better diagnose the logical flaw and produce a correct fix. Conversely, an incorrect oracle can mislead the debugging process, causing the LLM to alter correct code or introduce new bugs. This end-to-end evaluation confirms that the improvements in oracle generation from Nexus are not merely statistical but yield tangible benefits for critical software engineering automation tasks.

## 6.3 IMPLICATIONS FOR RESEARCHERS AND DEVELOPERS

Our findings offer significant implications for both the academic research community and software development practitioners, charting a new course for LLM-driven software testing. For researchers, our work challenges the prevailing paradigm of single-paradigm, specification-based deliberation-only multi-agent systems in software engineering. The superior performance of Nexus demonstrates that structured deliberation among agents with diverse, orthogonal reasoning strategies, followed by validation and self-refinement, is more effective than seeking agreement from agents shar-

Table 7: Self-Debugging Performance (%), measured by the final Pass@1 after one round of debugging using feedback from the generated oracles. Deltas are shown relative to the Direct Generation baseline for each model.

| Method | HE | HE-Pro | MBPP | MBPP-Pro | TE | LCB | ULT |
|---|---|---|---|---|---|---|---|
| **Underlying Model: GPT-4.1-Nano** | | | | | | | |
| Direct Gen. Feedback | 29.55 | 9.27 | 13.25 | 11.18 | 7.04 | 1.00 | 37.70 |
| CANDOR Feedback | 20.45 (-9.10) | 2.42 (-6.85) | 8.20 (-5.05) | 20.57 (+9.39) | 21.48 (+14.44) | 4.98 (+3.98) | 37.70 (+0.00) |
| Nexus Feedback | **35.23 (+5.68)** | **14.52 (+5.25)** | **15.77 (+2.52)** | **20.69 (+9.51)** | **29.63 (+22.59)** | **11.30 (+10.30)** | **39.34 (+1.64)** |
| **Underlying Model: GPT-4.1-Mini** | | | | | | | |
| Direct Gen. Feedback | 28.41 | 7.66 | 5.68 | 16.41 | 18.52 | 16.28 | 34.43 |
| CANDOR Feedback | 35.23 (+6.82) | 11.29 (+3.63) | 14.83 (+9.15) | 20.33 (+3.92) | 34.81 (+16.29) | 15.61 (-0.67) | 32.24 (-2.19) |
| Nexus Feedback | **69.32 (+40.91)** | **16.53 (+8.87)** | **17.35 (+11.67)** | **22.71 (+6.30)** | **35.19 (+16.67)** | **18.60 (+2.32)** | **38.80 (+4.37)** |

ing a single epistemological foundation. This suggests a critical need to move beyond simple single-paradigm, deliberation-only mechanisms to explore more sophisticated agent interaction protocols, such as argumentation, debate, and evidence-based synthesis.

For developers and practitioners, our results signal a practical path toward more reliable automated testing pipelines. The success of the diverse panel of agents highlights the immediate value of incorporating multiple testing perspectives into the generation process, rather than relying solely on specifications. Our framework should be viewed not as a replacement for human testers but as a powerful augmentation tool. The final, validated oracles produced by Nexus can significantly accelerate the code review process by helping developers quickly verify complex test cases. Finally, our work reaffirms the foundational importance of high-quality documentation. The effectiveness of the entire pipeline still relies on an initial, clear specification, reminding practitioners that human-written documentation remains a critical component in the age of LLM-driven development. By leveraging frameworks like Nexus, development teams can enhance their bug detection capabilities and streamline the debugging cycle, ultimately leading to more robust and reliable software.

# 7 CONCLUSION

In this paper, we introduced Nexus, a novel multi-agent framework that establishes a new paradigm by grounding language-based reasoning in executable evidence. The core innovation of Nexus lies in its sequential pipeline of multi-perspective deliberation, validation against a candidate implementation, and iterative self-refinement based on execution feedback. Our extensive evaluation across seven diverse benchmarks and four LLMs demonstrates that Nexus consistently and substantially outperforms existing approaches. For instance, when powered by *GPT-4.1-Mini*, Nexus boosts test-level oracle accuracy on the challenging LiveCodeBench benchmark to $57.73\%$ from the state-of-the-art's $46.30\%$ achieved by CANDOR. Our evaluation results also demonstrate that the improvements are not only in oracle accuracy but also in the utility of these oracles for critical downstream tasks. Notably, the enhanced oracles from Nexus led to a significantly higher bug detection rate and more than doubled the success rate of automated program repair in our experiments. For example, test cases equipped with Nexus oracles successfully identify $95.45\%$ of buggy programs on the HumanEval benchmark, significantly outperforming both Direct Generation ($60.23\%$) and CANDOR ($90.91\%$). Furthermore, the high-fidelity feedback from Nexus's oracles proves transformative for downstream tasks, more than doubling the program repair rate in an automated debugging scenario to $69.32\%$ compared to the $35.23\%$ achieved by CANDOR.

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

APPENDIX

# A  ADDITIONAL RELATED WORK

## A.1  LLM FOR TEST INPUT GENERATION

Most of the test input generation works in the software testing community commonly referred to as test case generation, but in fact, more accurately characterized as test input generation accompanied by regression oracles during the regression testing phase. During the regression testing phase, these works in test input generation are primarily designed to automate the manual testing process by producing test inputs with high code coverage and mutation score. Traditional techniques in this area include fuzzing (Fioraldi et al., 2023; Miller et al., 1990), feedback-directed random test generation (Arteca et al., 2022; Csallner & Smaragdakis, 2004; Pacheco & Ernst, 2007; Pacheco et al., 2008; Selakovic et al., 2018), dynamic symbolic execution (Cadar et al., 2008; Godefroid et al., 2005; Sen et al., 2005; Tillmann et al., 2014), search-based approaches (Fraser & Arcuri, 2011a;b), and LLM-based approaches (Du et al., 2024; Qing et al., 2025; Huang et al., 2024). Among these approaches, EvoSuite (Fraser & Arcuri, 2011b) has been extensively studied and applied to more than a hundred open-source software projects and several industrial systems, identifying thousands of potential bugs. However, as pointed out by Zhang et al. (2025a), traditional approaches like EvoSuite often struggle to produce meaningful and human-readable test cases, despite their remarkable performance in code coverage. This is primarily due to their limited understanding of the semantics of the focal methods. To alleviate such issues, recent research has turned to LLMs to produce more practical test inputs with high readability and coverage. Tufano et al. (2021) introduced the first LLM-based unit test generation approach AthenaTest, which formulates the task as a sequence-to-sequence learning problem. AthenaTest first denoises the pre-training of LLMs on a large Java corpus and then performs supervised fine-tuning for the downstream task of test generation. This work inspired subsequent research on fine-tuning LLMs for test generation (Alagarsamy et al., 2023; He et al., 2024; Rao et al., 2023). Additionally, researchers have explored various strategies to improve the training process of LLMs in test case generation, including reinforcement learning Steenhoek et al. (2025), domain adaptation (Shin et al., 2024), and data augmentation (He et al., 2025). Alternatively, a myriad of works perform prompt engineering using off-the-shelf LLMs, following a generation-and-refinement paradigm, where initial test cases are first generated based on prompts and then iteratively refined using dynamic execution feedback (e.g., code coverage report and failure information) (Pizzorno & Berger, 2024; Gu et al., 2025; Ni et al., 2024; Ryan et al., 2024; Schäfer et al., 2023; Wang et al., 2024; Yuan et al., 2024; Zhang et al., 2025a). To further improve the quality of test inputs, researchers propose a wide range of strategies to incorporate LLMs with valuable contextual information, including mutation testing (Dakhel et al., 2023), method slicing (Wang et al., 2024), demonstration retrieval (Zhang et al., 2024), defect detection (Yin et al., 2024), and program analysis (Yang et al., 2025).

The automated generation of test inputs has been a central focus of software testing research for decades, driven by the goal of achieving high code coverage while minimizing manual effort. Early approaches relied on random testing and fuzzing techniques (Miller et al., 1990; Fioraldi et al., 2023), which generate inputs by introducing random mutations or following predefined patterns. While effective for discovering crashes and security vulnerabilities, these techniques often struggle to generate semantically meaningful inputs for complex functions. Feedback-directed random testing (Pacheco & Ernst, 2007; Pacheco et al., 2008) improved upon pure random approaches by incorporating runtime feedback to guide input generation toward unexplored paths. Tools like Randoop (Pacheco & Ernst, 2007) demonstrated that lightweight dynamic analysis could significantly improve test effectiveness while maintaining the simplicity of random generation.

Dynamic symbolic execution emerged as a more systematic approach to test input generation, combining concrete execution with symbolic reasoning to explore program paths methodically (Cadar et al., 2008; Godefroid et al., 2005; Sen et al., 2005). By maintaining symbolic constraints along execution paths and solving them with SMT solvers, these techniques can generate inputs that exercise specific branches and statements. However, symbolic execution faces scalability challenges with complex programs, particularly those involving loops, recursion, and external dependencies (Tillmann et al., 2014). Search-based software testing approaches, exemplified by tools like EvoSuite (Fraser & Arcuri, 2011b), frame test generation as an optimization problem. Using evolu-

tionary algorithms guided by fitness functions based on coverage metrics, these tools have achieved remarkable success in generating high-coverage test suites for object-oriented programs (Fraser & Arcuri, 2011a). EvoSuite has been applied to hundreds of real-world projects, demonstrating the practical viability of search-based approaches.

The advent of large language models has introduced a paradigm shift in test input generation. Early work by Tufano et al. (2021) demonstrated that transformer-based models could learn to generate meaningful test cases from code context. This inspired a wave of research exploring different training strategies, including reinforcement learning (Steenhoek et al., 2025), domain adaptation (Shin et al., 2024), and data augmentation (He et al., 2025). Contemporary LLM-based approaches typically follow a generation-and-refinement paradigm, where initial test cases are iteratively improved using execution feedback (Pizzorno & Berger, 2024; Gu et al., 2025; Ni et al., 2024). These methods leverage various forms of context to enhance generation quality, including method slicing (Wang et al., 2024), demonstration retrieval (Zhang et al., 2024), and program analysis (Yang et al., 2025). While these advances have significantly improved test input generation, they predominantly rely on regression oracles that assume the correctness of existing implementations, limiting their ability to detect functional bugs.

## A.2 LLM FOR TEST ORACLE GENERATION

The generation of test oracles presents fundamentally different challenges from input generation, requiring a deep understanding of intended program behavior rather than mere code coverage. Early automated testing tools largely sidestepped this challenge by employing regression oracles or simple crash oracles (Fraser & Arcuri, 2011b; Pacheco & Ernst, 2007). While regression oracles effectively detect behavioral changes between versions, they cannot validate functional correctness against specifications, potentially perpetuating bugs across software evolution (Pizzorno & Berger, 2024). The specification oracle problem that generating oracles that verify correctness against intended behavior has long been recognized as one of the most challenging aspects of automated testing (Barr et al., 2014). The emergence of LLM-based approaches brought new possibilities for oracle generation. TOGA (Dinella et al., 2022) pioneered the use of transformer models for inferring specification-based oracles, fine-tuning CodeBERT to predict expected outputs from code context. This work inspired subsequent research exploring various neural architectures and training strategies for oracle generation (Endres et al., 2024; Hayet et al., 2024). TOGLL (Hossain & Dwyer, 2024) achieved state-of-the-art performance by combining EvoSuite-generated inputs with fine-tuned LLM oracles, demonstrating that hybrid approaches could leverage the strengths of both traditional and learning-based methods. However, fine-tuning approaches face significant limitations in generalization, as model performance degrades substantially when test distributions shift from training data (Xu et al., 2025). Recent work has increasingly focused on prompt engineering strategies that leverage the zero-shot capabilities of LLMs. Siddiq et al. (2024) demonstrated that carefully crafted prompts could enable LLMs to generate both inputs and oracles from natural language descriptions without fine-tuning. RetriGen (Zhang et al., 2025b) enhanced oracle generation through retrieval-augmented generation, incorporating relevant assertions from external codebases to guide prediction. CANDOR (Xu et al., 2025) represents the current SOTA method, which introduces a multi-agent framework where several LLM agents collaborate through a panel discussion to reach a consensus on the correct oracle. While CANDOR achieves impressive results without fine-tuning, our analysis reveals that its single-paradigm and deliberation-only approach limits its ability to detect bugs beyond explicit specifications. In this work, we propose `Nexus`, which overcomes these limitations by introducing our novel deliberation-validation-refinement architecture. `Nexus` employs a multi-agent framework that facilitates diverse reasoning strategies, enhancing the initial analysis of test oracles. By incorporating a validation phase that grounds oracles in executable contexts and a self-refinement loop that leverages runtime feedback, `Nexus` further improves the accuracy of generated oracles.

## B ROBUSTNESS ANALYSIS OF EXECUTION-BASED VALIDATION

A critical design choice in `Nexus` is adapting the tentative oracles based on the candidate implementation, rather than adapting the code based on oracles. This decision aligns with our primary objective of test oracle generation. Analogous to self-refinement in code generation—where tests verify

code—we invert the relationship, using the candidate implementation (treated as a high-confidence proxy) to verify the oracle.

To empirically validate this approach, we analyzed the validation phase outcomes on the HumanEval benchmark using *GPT-4.1-Nano*. We categorized the validation results into four scenarios based on the ground truth correctness of both the oracle and the candidate implementation, as shown in Tab. 8.

Table 8: Analysis of Validation Phase Scenarios on HumanEval (GPT-4.1-Nano). We analyze cases where validation fails (oracle mismatch with candidate implementation).

| Scenario | Percentage |
|---|---|
| A: True Negatives (Oracle Incorrect, Candidate Correct) | 6.5% |
| B: Successful Correction (subset of A) | 88.3% |
| C: False Negatives (Oracle Correct, Candidate Incorrect) | 16.6% |
| D: Successful Recovery (subset of C) | 55.4% |

The analysis reveals that in 6.5% of cases (Scenario A), the validation correctly identified an incorrect oracle (True Negative), and the self-refinement loop successfully corrected 88.3% of these (Scenario B). Crucially, even when the candidate implementation was incorrect (Scenario C, 16.6% of cases), leading to a "false negative" validation signal, the self-refinement loop demonstrated remarkable resilience. In 55.4% of these cases (Scenario D), the oracle remained correct after refinement. This suggests that the model can distinguish between genuine oracle errors and implementation mismatches during the self-refinement process, preventing valid oracles from being corrupted by incorrect candidate code.

## C    ADDITIONAL ABLATION STUDY ON ROLE SPECIALIZATION

Table 9: Comparison of removing vs. replacing specialized agents with a generic agent. Results are reported for HumanEval (HE) and HumanEval-Pro (HE-Pro).

| Setting | HE (Remove) | HE (Replace) | HE-Pro (Remove) | HE-Pro (Replace) |
|---|---|---|---|---|
| NEXUS (Full) | 93.69 | | 79.47 | |
| w/o SE | 91.62 | 92.47 | 73.47 | 78.36 |
| w/o ES | 91.22 | 92.41 | 74.18 | 78.33 |
| w/o FV | 90.88 | 92.29 | 74.24 | 78.36 |
| w/o AA | 89.36 | 92.74 | 72.45 | 77.80 |

## D    COMPARISON WITH RELATED WORK AND EVALUATION DIVERSITY

To address the need for comparison with related work and evaluation diversity, we conducted a new experiment on TestGenEval (Jain et al., 2024a). We chose TestGenEval as it is built from SWE-Bench (addressing the need for more realistic benchmarks) and uses a pass@1 metric that aligns well with our evaluation. Given the high computational cost of running this benchmark, we evaluated all frameworks on a randomly selected subset of 164 tasks. We also included comparisons to strong, relevant baselines like SWT-Agent (Mündler et al., 2024) and Otter (Ahmed et al., 2025). The results in Tab. 10 demonstrate that Nexus outperforms all baselines, including those specifically designed for SWE-type tasks. This provides strong evidence that our framework's approach is not limited to academic benchmarks and generalizes effectively to more complex, real-world test generation scenarios.

## E    ADDITIONAL EVALUATION ON MODEL DIVERSITY

The choice of LLMs in our main evaluation involved analyzing their reasoning capabilities, cost, and common usage. We prioritized reasoning-capable models or those with large parameter counts

Table 10: Pass@1 (%) on TestGenEval.

| Method | Pass@1 (%) |
|--------|------------|
| GPT-4.1-Nano | 50.6 |
| CANDOR | 62.8 |
| SWT-Agent | 54.3 |
| Otter | 65.2 |
| **Nexus** | **67.1** |

suitable for agentic frameworks. Specifically, we selected GPT-4.1-Nano and GPT-4.1-Mini as base LLMs for their balance of performance and cost-effectiveness.

To address concerns regarding evaluation diversity and demonstrate the generalizability of `Nexus`, we conducted additional experiments on two challenging benchmarks, TestEval and Live-CodeBench, using two other LLMs: GPT-4.1 and GPT-4-Sonnet. As presented in Tab. 11, `Nexus` consistently improves performance across these diverse models. For GPT-4.1, `Nexus` improves accuracy by 3.9% on TestEval and 12.9% on LiveCodeBench. For GPT-4-Sonnet, the improvements are even more pronounced, with gains of 25.4% on TestEval and 13.1% on LiveCodeBench. These results confirm that the benefits of the `Nexus` framework can extend to a wider range of models beyond the open-source models evaluated.

Table 11: Performance comparison on TestEval and LiveCodeBench with additional LLMs.

| Model | TestEval | LiveCodeBench |
|-------|----------|---------------|
| GPT-4.1 | 66.8 | 61.3 |
| +Nexus | 70.7 (+3.9) | 74.2 (+12.9) |
| GPT-4-Sonnet | 56.7 | 43.4 |
| +Nexus | 82.1 (+25.4) | 56.6 (+13.1) |

# F    COST ANALYSIS

`Nexus` intentionally trades computation for higher accuracy. To quantify this trade-off, we provide the detailed cost analysis for the HumanEval evaluation using *GPT-4.1-Nano* for each framework with the average token usage, runtime, and estimated API cost in Tab. 12.

Table 12: Computational cost of `Nexus` and baselines.

| Framework | Avg. Tokens (M) | Avg. Runtime (s) | Est. API Cost ($) |
|-----------|-----------------|------------------|-------------------|
| Direct Generation | 0.2 | 15.42 | 0.06 |
| CANDOR | 3.5 | 67.28 | 1.06 |
| Nexus (Ours) | 4.8 | 87.63 | 1.42 |

# G    NEXUS PROMPTS

This section provides the detailed prompts used to instruct the LLMs for each agent role within the `Nexus` framework. Each prompt is designed to elicit specific reasoning patterns and behaviors crucial for the agent's function in the pipeline. For the **Direct Generation** baseline, the prompt is identical to the *Tentative Oracle Generator* used in `Nexus`, with a key difference in the system prompt. The instruction "Be quick, but it may not be perfect" is removed, as it is specific to the tentative nature of the initial oracles in `Nexus` (a setup adopted from CANDOR to limit excessive token consumption by reasoning LLMs during this phase).

### G.1 TENTATIVE ORACLE GENERATOR

**System Prompt**

```
You are generating initial test oracles.
Be quick, but it may not be perfect.
```

**Prompt**

```
Generate test assertions for the following function:

Task Description:
{task_description}

Test inputs:
{test_inputs_formatted}

You MUST provide EXACTLY {len(test_inputs)} assertions:
```

### G.2 DIRECT GENERATION BASELINE

**System Prompt**

```
You are generating initial test oracles.
```

**Prompt**

```
Generate test assertions for the following function:

Task Description:
{task_description}

Test inputs:
{test_inputs_formatted}

You MUST provide EXACTLY {len(test_inputs)} assertions:
```

### G.3 REQUIREMENT ENGINEER

**System Prompt**

```
You are an expert software engineer and requirement analyst.
Extract requirements and generate clear specifications in predicate
logic when possible.
```

**Prompt**

```
Analyze the following function specification and extract key functional
requirements:
{task_description}
```

### G.4 PANELISTS

**System Prompt (Common to all Panelists)**

```
You are a Senior Software Engineer specializing in testing. Analyze
the test oracles generated by test generation tools, which may not be
perfect. Identify incorrect test oracles and provide corrected versions
based on the requirements.
```

**Prompt Template**

```
You are analyzing test oracles generated by an automated tool.
Your role: {role_name} - {role_focus}

Description:
{task_description}

Requirements:
{requirements[task_idx]}

Test Cases and Current Oracles:
{tentative_formatted}

Test Inputs:
{test_inputs_formatted}

Output your final {len(test_inputs)} test assertions in this exact format:

```python
assert function_name(input1) == expected_output1
assert function_name(input2) == expected_output2
````

Each assertion on one line, no additional code.
```

**Panelist Roles** The four roles assigned to the panelists are:

- **Specification Expert:** Focuses on adherence to documented specifications and requirements.
- **Edge Case Specialist:** Focuses on boundary conditions, corner cases, and error scenarios.
- **Functional Validator:** Focuses on core functionality and expected input-output relationships.
- **Algorithmic Analyst:** Focuses on step-by-step algorithm execution and correctness.

G.5 INTERPRETER

**System Prompt**

```
You work with an excellent software tester who is trying to check if
test cases have correct test oracles. The tester always gets correct
oracles but lacks confidence and overthinks. Your task is to summarize
the tester's thoughts and extract the correct oracles from the analysis.
```

**Prompt**

```
A software tester has analyzed test oracles and provided detailed thoughts.
The tester is excellent but lacks confidence and tends to overthink.
Your job is to extract the key insights and conclusions.

Tester's Thoughts:
{panelist_output}

Test Code Being Analyzed:
{test_code_formatted}
```

G.6 CURATOR

**System Prompt**

```
You are a senior software engineer managing a team of software testers.
Your team has analyzed test cases and provided analysis reports. Summarize
their analysis and provide final judgment on the test oracles, making
corrections where necessary.
```

**Prompt**

```
You are managing a team of software testers analyzing test cases generated
by a competitor.
Three team members have analyzed the test oracles and provided their reports.

Task Description:
{task_description}

Current Test Oracles:
{tentative_formatted}

Test Inputs:
{test_inputs_formatted}

Team Members' Analysis Reports:
{panel_discussion}

Output your final {len(test_inputs)} test assertions in this exact format:

'''python
assert function_name(input1) == expected_output1
assert function_name(input2) == expected_output2
'''

Each assertion on one line, no additional code.
```

## G.7  CANDIDATE CODE GENERATOR

Note that the test examples provided in the prompt are extracted from the original task description.

**System Prompt**

```
You are an expert Python programmer. Generate clean, correct, and
efficient function implementations based on specifications.
```

**Prompt**

```
Generate a Python function implementation based on the following
specification:

Task Description:
{task_description}

Function Name: {function_name}

Example Usage:
{test_examples}

Requirements:

1.  Implement ONLY the function, no additional code
2.  Function should be complete and runnable
3.  Handle edge cases appropriately
```

4.  Follow the exact function signature from the examples

Provide ONLY the Python function implementation:

### G.8   SELF-REFINEMENT

**System Prompt**

You are an expert debugger specializing in test oracle correction.
Analyze ALL failed assertions together, understand the errors, and
provide precise fixes for each. Always provide syntactically correct
Python assertions.

**Prompt**

Fix ALL the following failed test oracles for the given function.
{iteration_note}

Task Description:
{task_description}

Function Implementation Being Tested:

```python
{candidate_code}
```

{passed_examples}

ALL Failed Oracles to Fix:
{failed_oracles_formatted}

Analyze why each oracle failed and provide corrected versions for ALL
of them.
Consider:

1.  The expected output type and format based on the function implementation
2.  Edge cases and boundary conditions
3.  The error messages for each failed oracle
4.  Common patterns from the passing oracles
    {f"5. Why the previous iteration's fixes didn't work (iteration
    {iteration + 1})" if iteration > 0 else ""}

Output your final {len(failed_oracles)} test assertions in this exact
format:

```python
assert function_name(input1) == expected_output1
assert function_name(input2) == expected_output2
```

Each assertion on one line, no additional code.

