# OpenReview forum: "Nexus: Execution-Grounded Multi-Agent Test Oracle Synthesis"
_ICLR.cc/2026/Conference — Submitted to ICLR 2026_

### Official Review · Reviewer_zCg8 · 2025-10-26

**Soundness:** 3
**Presentation:** 2
**Contribution:** 2
**Rating:** 4
**Confidence:** 3

**Summary:**

This work introduces Nexus, a multi-agent framework for automated test oracle generation. Nexus operates through three phases, namely, deliberation, validation, and iterative self-refinement. First, four specialized agents in the delibration phase collaboratively generate and critique candidate oracles, then the validation phase validates them by executing against an LLM-generated implementation in a secure environment. In the end, the self-refinement phase iteratively refines those oracles that fail. Nexus is evaluated on seven benchmarks and four LLMs (both open- and close-source), performing better in oracle accuracy, bug detection, and automated program repair success. For example, it improves the test-level accuracy on LiveCodeBench from 46.3% to 57.7% and doubling repair success rates.

**Strengths:**

- Multi-agent approach and splitting the problem into multiple sub-problems
- Effectiveness compared to existing approaches

**Weaknesses:**

- The validation phase relies on LLM-generated plausible implementation. Why someone should trust on this implementation as ground-truth?
- All benchmarks are crafted and the size of programs are very small. For example, HumanEval.
- This architecture can be very expensive in reality, both in terms of dollar and runtime cost. No discussions about this in the paper.

**Questions:**

- Why the LLM-generated implementation should be trusted in the validation phase?
- What is the cost of Nexus? What is the token usage, total time in seconds of each component, dollar amount?
- Why the authors did not evaluate on real-life projects, with large codebases?

---

> ### Author Response · Authors · 2025-11-19
>
> **Q1&W1: Why the LLM-generated implementation should be trusted in the validation phase?**
>
> This is a good question. First, the key reason for us to use the LLM-generated code to adapt the tentative oracle, but not the code itself, is that this work **focuses on test oracle generation**. Specifically, similar to existing works on self-refine/debug-based code generation, which first generate candidate tests and then use these test cases to debug and refine the LLM-generated code, we transfer the optimization object from the code to the oracles. Next, we do not treat the LLM-generated code as fully correct (just as code generation works would not treat tests generated by LLMs as fully correct). We treat the LLM-generated code as having high confidence in its correctness, and then we use this code to further check the correctness of the generated oracle. Furthermore, as our experiments show, even when the provided code solution is incorrect, LLMs still have the ability to ensure that a correct test oracle (which may initially fail due to the incorrect code solution) is preserved after the self-refine stage. Finally, for the self-refine component in Nexus, as our results in Table 1 demonstrate that it can recover some incorrect oracles to correct ones, we believe that this component has value. To address the reviewer’s concern, we will provide an **argparse** setting in the final code to allow users to decide whether to use this self-refine component.
>
> | **Scenario** | **Count / %** |
> |--|--|
> | A: **True Negatives** (Oracle was *Incorrect*, Candidate was *Correct*) | 6.5% |
> | B: Successful *Correction* (Refinement loop *fixes* the incorrect oracle) | 88.3% (of A) |
> | C: **False Negatives** (Oracle was *Correct*, Candidate was *Incorrect*) | 16.6% |
> | D: Successful *Recovery* (Refinement loop *preserves* the correct oracle) | 55.4% (of C) |
>
> *Table 1: Analysis of Oracle's Failing Validation Phase (HumanEval, GPT-4.1-Nano)*
>
> **W3&Q2: What is the cost of Nexus? What is the token usage, total time in seconds of each component, dollar amount?**
>
> Good question! Nexus intentionally trades computation for higher accuracy. To quantify this trade-off, we provide the detailed cost analysis for the HumanEval evaluation using GPT-4.1-Nano for each framework with the average token usage, runtime, and estimated API cost below:
>
> | **Framework** | **Avg. Tokens (M)** | **Avg. Runtime (s)** | **Est. API Cost ($0.15/1M in, $0.60/1M out)** |
> | :--- | :---: | :---: | :---: |
> | Direct Generation | 0.2 | 15.42 | $0.06 |
> | CANDOR | 3.5 | 67.28 | $1.06 |
> | Nexus (Ours) | 4.8 | 87.63 | $1.42 |
>
> *Table 2: Computational cost of Nexus and baselines.*
>
> **W2&Q3: Why the authors did not evaluate on real-life projects, with large codebases?**
>
> Thank you for this valuable suggestion. Following your suggestion, we conducted a new experiment on TestGenEval [1]. We chose TestGenEval as it is built from SWE-Bench (addressing the reviewer's request for more realistic benchmarks) and uses a pass@1 metric that aligns well with our evaluation. Given the high computational cost of running this benchmark, we evaluated all frameworks on a randomly selected subset of 164 tasks. As requested, we also included comparisons to strong, relevant baselines like SWT-Agent [2] and Otter [3]. The results in Table 4 demonstrate that Nexus outperforms all baselines, including those specifically designed for SWE-type tasks. This provides strong evidence that our framework's approach is not limited to academic benchmarks and generalizes effectively to more complex, real-world test generation scenarios.
>
> | Method | Pass@1 (%) |
> |--|--|
> | GPT-4.1-Nano | 50.6 |
> | Candor | 62.8 |
> | SWT-Agent | 54.3 |
> | Otter | 65.2 |
> | Nexus | 67.1 |
>
> *Table 4: Pass@1 (%) on TestGenEval.*
>
> [1] Jain, Kush, Gabriel Synnaeve, and Baptiste Roziere. "Testgeneval: A real world unit test generation and test completion benchmark." arXiv preprint arXiv:2410.00752 (2024).
>
> [2] Mündler N, Müller M, He J, et al. SWT-bench: Testing and validating real-world bug-fixes with code agents[J]. Advances in Neural Information Processing Systems, 2024, 37: 81857-81887.
>
> [3] Ahmed, T., Ganhotra, J., Pan, R., Shinnar, A., Sinha, S., & Hirzel, M. (2025). Otter: Generating Tests from Issues to Validate SWE Patches. arXiv preprint arXiv:2502.05368.

---

> > ### Comment · Reviewer_zCg8 · 2025-11-23
> >
> > Thank you for answering my questions. I have increased my score.

---

> > > ### Author Response · Authors · 2025-11-24
> > >
> > > Thank you for your appreciation and for increasing your score. We will ensure that all points discussed in the rebuttal are incorporated into the final version of the paper.

---

### Official Review · Reviewer_bF44 · 2025-10-27

**Soundness:** 3
**Presentation:** 3
**Contribution:** 2
**Rating:** 2
**Confidence:** 4

**Summary:**

The paper presents Nexus, a multi-agent framework for automatically generating test oracles for individual functions. The system operates in three stages: (1) multi-agent deliberation among specialized testers; (2) validation through sandboxed execution; and (3) self-refinement using runtime error feedback. Experiments show improvements on standard benchmarks—e.g., oracle accuracy on LiveCodeBench rising from 46.3% to 57.7% (GPT-4.1-Mini) and bug detection on HumanEval improving modestly.

**Strengths:**

- The multi-agent deliberation and self-refinement pipeline is well designed and demonstrates good engineering effort.
- Clear and consistent improvements across multiple benchmarks.
- The validation loop leveraging execution feedback is a reasonable and practical idea.

**Weaknesses:**

1. Fundamental Paradox: Solves Only the Easy Problems

Nexus performs well only on simple, well-specified functions where expected behavior is trivial. Ironically, these are exactly the cases where human developers can easily write tests themselves, while the harder, more ambiguous cases remain out of reach.

2. Severely Limited Applicability (Technical Scope)

The framework assumes isolated, stateless, pure functions with deterministic I/O. It cannot handle realistic challenges such as shared state, object interactions, asynchronous behavior, or dependencies on databases, APIs, or file systems. In real software, these assumptions rarely hold, making the current design applicable to only a small subset of cases.

3. Circular Reasoning and Validation Ambiguity

Both the oracle generation and validation phases rely on model-generated reasoning. If the candidate implementation and oracle share the same misunderstanding, the validation step can falsely confirm incorrect behavior. Without an external specification or reference, this becomes a self-reinforcing evaluation loop.

4. Questionable Real-World Scenarios (Practical Value)

Even if technically sound, it’s unclear when developers would realistically use Nexus.
- For new code, they already know the intended behavior and can write tests faster.
- For legacy code, there’s no reliable ground truth for Nexus to infer correct behavior.
- For integration or regression testing, the system’s single-function assumption breaks down.

Thus, while academically interesting, the method lacks a clear use case in everyday software development.

**Questions:**

N/A

---

> ### Author Response · Authors · 2025-11-19
>
> **W1 and W2: Fundamental Paradox**
>
> We clarify that, first, Nexus not only works for well-specified functions, but it can also work for real-world functions (e.g., UTL) and even SWE tasks. Specifically, to address the reviewer’s concern, we provide the evaluation results of Nexus and baselines on TestGenEval [1], which is constructed from SWE-Bench and used to measure the LLM's performance in constructing test cases for SWE tasks. The evaluation results are shown in Table 1, where we can observe that Nexus can improve the accuracy of LLM-generated test cases for SWE tasks from 50.6% to 67.1%.
>
> | Method | Pass@1 (%) |
> |--|--|
> | GPT-4.1-Nano | 50.6 |
> | Candor | 62.8 |
> | SWT-Agent | 54.3 |
> | Otter | 65.2 |
> | Nexus | 67.1 |
>
> *Table 1: Pass@1 (%) on TestGenEval.*
>
> **W3: Circular Reasoning and Validation Ambiguity: Both the oracle generation and validation phases rely on model-generated reasoning. If the candidate implementation and oracle share the same misunderstanding, the validation step can falsely confirm incorrect behavior. Without an external specification or reference, this becomes a self-reinforcing evaluation loop.**
>
> This is a good question. First, the key reason for us to use the LLM-generated code to adapt the tentative oracle, but not the code itself, is that this work **focuses on test oracle generation**. Specifically, similar to existing works on self-refine/debug-based code generation, which first generate candidate tests and then use these test cases to debug and refine the LLM-generated code, we transfer the optimization object from the code to the oracles. Next, we do not treat the LLM-generated code as fully correct (just as code generation works would not treat tests generated by LLMs as fully correct). We treat the LLM-generated code as having high confidence in its correctness, and then we use this code to further check the correctness of the generated oracle. Furthermore, as our experiments show (see the case study and Table 2), even when the provided code solution is incorrect, LLMs still have the ability to ensure that a correct test oracle (which may initially fail due to the incorrect code solution) is preserved after the self-refine stage. Finally, for the self-refine component in Nexus, as our results in Table 2 demonstrate that it can recover some incorrect oracles to correct ones, we believe that this component has value. To address the reviewer’s concern, we will provide an **argparse** setting in the final code to allow users to decide whether to use this self-refine component.
>
> | **Scenario** | **Count / %** |
> |--|--|
> | A: **True Negatives** (Oracle was *Incorrect*, Candidate was *Correct*) | 6.5% |
> | B: Successful *Correction* (Refinement loop *fixes* the incorrect oracle) | 88.3% (of A) |
> | C: **False Negatives** (Oracle was *Correct*, Candidate was *Incorrect*) | 16.6% |
> | D: Successful *Recovery* (Refinement loop *preserves* the correct oracle) | 55.4% (of C) |
>
> *Table 2: Analysis of Oracle's Failing Validation Phase (HumanEval, GPT-4.1-Nano)*
>
> **W4: Questionable Real-World Scenarios (Practical Value)**
>
> Good question! We clarify that, first, as mentioned in the response for W1&W2, Nexus can also work for SWE-level projects to construct test cases, which can reduce the manual effort of software developers. Next, regarding the usefulness of Nexus in the three scenarios proposed by the authors, one commonly used scenario is for new code. Specifically, as mentioned by existing works on test oracle generation [1-3], we agree that human developers can write tests for the code when they know the intended behaviors. However, a key reason for existing works on test oracle generation [1-7] is to try to reduce the human effort in the test construction process, which takes up almost half of the time for developers in the software life cycle for development and maintenance.
>
> [1] Ahmed T, Ganhotra J, Pan R, et al. Otter: Generating Tests from Issues to Validate SWE Patches[J]. arXiv preprint arXiv:2502.05368, 2025.
>
> [2] Xu Q, Wang G, Briand L, et al. A Multi-agent LLM-based JUit Test Generation with Strong Oracles[J]. arXiv preprint arXiv:2506.02943, 2025.
>
> [3] Mündler, Niels, et al. "SWT-bench: Testing and validating real-world bug-fixes with code agents." Advances in Neural Information Processing Systems 37 (2024): 81857-81887.
>
> [4] Jain, Kush, Gabriel Synnaeve, and Baptiste Roziere. "Testgeneval: A real world unit test generation and test completion benchmark." arXiv preprint arXiv:2410.00752 (2024).
>
> [5] Brooks Jr, Frederick P. The mythical man-month: essays on software engineering. Pearson Education, 1995.
>
> [6] Planning, Strategic. "The economic impacts of inadequate infrastructure for software testing." National Institute of Standards and Technology 1.2002 (2002): 1.
>
> [7] Barr, Earl T., et al. "The oracle problem in software testing: A survey." IEEE transactions on software engineering 41.5 (2014): 507-525.

---

> > ### Comment · Reviewer_bF44 · 2025-11-26
> >
> > Thank you for the detailed rebuttal.
> >
> > While the additional experiments are helpful, several core concerns remain. In real SWE practice, test construction is typically guided by specifications, code review, and documented requirements, and tests at the module or integration level are not derived through unconstrained reasoning. As a result, relying on LLM-generated oracles and LLM-generated code for mutual validation raises questions about robustness and realism. The added TestGenEval results do not fully address this gap, as they do not reflect the complexity or workflows of actual CI pipelines or production-grade test suites.
> >
> > Overall, although the clarifications are appreciated, I believe the paper would benefit from further evidence and refinement before it can convincingly support its claims in real-world SWE settings.

---

> > > ### Author Response · Authors · 2025-11-27
> > >
> > > We thank the reviewer for the continued engagement and the thoughtful critique regarding the robustness and realism of mutual validation in real-world SWE settings. We believe there may be a slight misalignment regarding the research context of our work compared to manual industry practices. We would like to clarify how Nexus fits into the established literature of Automated Test Generation (ATG) and the Oracle Problem, distinct from manual verification workflows.
> > >
> > > **1. Contextualizing "Unconstrained Reasoning" vs. The Oracle Problem** The reviewer correctly notes that in manual SWE practice, tests are derived from explicit requirements and code reviews. However, in the field of Automated Test Generation, formal specifications are rarely available in machine-readable formats. As defined in the survey on the Oracle Problem [1], the challenge is precisely that automated tools lack the "ground truth" that human testers possess. Most automated tools (e.g., EvoSuite) rely on Regression Oracles (assuming the current implementation is correct) or Implicit Oracles (checking for crashes). These methods cannot detect functional logic errors in new code. Nexus targets this specific gap by acting as a Specification-Based Oracle Generator that extracts implicit specifications (from natural language docstrings) to verify code. While less robust than a human with a formal spec document, it is a significant advancement over the current automated standard. **To ensure this context is clear to future readers, we have provided detailed background on this distinction and the scope of our work in the revised "Background" subsection of the manuscript. We hope this clarification addresses your concern regarding the positioning of our approach.**
> > >
> > > **2. Robustness of Mutual Validation (N-Version Programming)** Regarding the concern about "circular reasoning" when relying on LLM-generated oracles and code: While we acknowledge the theoretical risk, our approach draws inspiration from N-Version Programming and Differential Testing. The premise is that while an LLM might make a mistake in generation (Code) or reasoning (Oracle), the probability of it making the exact same semantic error in both modalities simultaneously is significantly lower than making an error in just one. Our validation phase acts as a consistency check. As shown in our ablation study (Table 2 of the previous response), this mechanism successfully filters out 55.4% of false negatives and corrects 88.3% of incorrect oracles. This empirical evidence suggests that "mutual validation" provides a practical robustness check, even without an external formal specification.
> > >
> > > **3. Practical Value: Augmentation vs. Replacement** We agree that Nexus does not yet replicate the complexity of a full production-grade CI pipeline involving integration tests and database states. However, the scope of this work is Unit Testing, which remains the foundational layer of the testing pyramid. The immediate value of Nexus is not to replace human QA, but to augment developers during the "Test-First" or "Test-After" coding loops. By automatically synthesizing a draft of specification-based oracles that are grounded in execution, Nexus reduces the "cold start" burden for developers, allowing them to review and refine tests rather than writing them from scratch.
> > >
> > > We hope this clarifies that Nexus is designed to solve the Oracle Problem in automated testing scenarios where formal specs are absent, i.e., a known hard problem in the literature, rather than attempting to replace the rigorous manual verification processes of critical systems. We believe the significant improvements over state-of-the-art baselines (CANDOR, etc.) on challenging benchmarks like TestGenEval and LiveCodeBench demonstrate a meaningful step forward in this specific research domain.
> > >
> > > Finally, we are committed to bridging the gap between our current evaluation and the "real-world" standards you emphasized. To ensure we address your concerns precisely in the final version, could you provide concrete examples of the specific benchmarks or CI/CD workflows (e.g., specific open-source repositories or integration suites) that you believe would conclusively demonstrate robustness in this context? While we believe our results on LiveCodeBench and TestGenEval are strong indicators of utility, understanding your specific acceptance criteria for "realism" would allow us to incorporate the exact evidence you require.
> > >
> > >
> > > [1] Barr, E. T., Harman, M., McMinn, P., Shahbaz, M., & Yoo, S. (2014). The oracle problem in software testing: A survey. IEEE transactions on software engineering, 41(5), 507-525.

---

### Official Review · Reviewer_6rvk · 2025-10-27

**Soundness:** 1
**Presentation:** 2
**Contribution:** 1
**Rating:** 2
**Confidence:** 4

**Summary:**

This work presents a method (Nexus), which uses a multi-agent framework and execution feedback to generate accurate test oracles for functions specified in natural language. The framework first generates test oracles one-shot using an LLM, then prompts an LLM with 4 different focus topics to find possible flaws, and aggregates the findings using an LLM. Then an LLM generates a candidate solution to the given function, and the output of that function is compared to the the generated oracles. In a self-refinement loops, an LLM can now adjust the generated oracles based on the execution feedback of the candidate solution. The method is evaluated on 8 different single-function benchmarks and with Qwen 7B/32B and GPT-4.1-nano/mini and shows between 1 and 40% improvements over a zero-shot baseline.

**Strengths:**

I think the idea of using specialized agents to analyze method outputs is a good idea to improve upon CANDOR. The figures and tables are legible.

**Weaknesses:**

I think this paper has several critical weaknesses, listed below.


**Lack of evidence**
It is unclear if the method results in true gains as claimed, and if its performance gains really stem from the proposed setup.
- Table 6 only shows performance difference after a round of debugging information after direct generation/CANDOR/Nexus. Please also include the performance without debugging information as a true baseline so that it can be seen whether the oracles increase performance at all.
- The method only allows adapting the tentative oracles based on execution feedback of the candidate solution, but not to adapt the candidate solution based on the generated oracles. This seems to presume that the candidate solution is correct - which clearly should not be assumed for a general task, and especially not if this method were to be used in a TDD-like setting as implied by the results presented in Table 6. Moreover, this raises the question why the candidate solution is not used directly to obtain adequate oracles, without the various discussions before. Does the candidate generation use the generated test oracles? This should generally be clarified in the text (and prompt B.6, e.g. what does test_examples refer to?)
- After all, the execution feedback seems to have marginal effect on the quality of the obtained oracles (Table 5). Given all the negative side effects of this execution feedback setup, I would recommend to majorly overhaul it or to drop it entirely  for a subsequent revision.
- The authors do not provide the prompt for the direct generation baseline. The prompt for the tentative oracle baseline (App B.1) directly tells the model to "[b]e quick, but it may not be perfect". Why is this included? I would expect this to (artificially) decrease the quality of generated tentative oracles, thus making the effect of the discussion seem to have a strong positive effect. Is the same prompt used for direct generation and how would the results look if the same prompt for direct generation was used for the tentative oracle generator?
- The ablation for different agents does just remove critics from the nexus setups (sec 5.2). I think it would be cleaner to replace them with a generically prompted agent, strengthening the claim that the improvements in Nexus stem from more specific/diverse prompts. Currently, it mainly shows that fewer critics/fewer inputs to aggregate over reduce the performance of nexus.
- Why exactly were the 4 agents chosen that nexus employs? Do they result from the preliminary study, or are they chosen based on a validation setting, or based on educated guesses by the authors on what might be useful? Were other critics tried?

**Lack of comparison to related work**
- The work only compares to a rudimentary baseline and only a single related work, and even there makes no strong case that exactly the proposed improvements result in the claimed superiority. The main comparison is to CANDOR, which is a single work that uses multi-agent debate, similarly to the presented work, but does not use different prompts to discuss tentative oracles. The authors claim they outperform CANDOR specifically due this difference (+ leveraging execution feedback) but don't make clear if their comparison implementation of CANDOR actually differs only in this regard (down to information flow, used prompts etc).
- Other baselines would help contextualize the results, e.g., comparison to any of the many methods outlined in Table 1 of [1] or any of the testing agents designed and evaluated for SWT-bench [2]
- In several places (e.g. Line 450) the authors highlight how prior work is all "single-paradigm" and "no-execution feedback". This is, from what I see, only based on a comparison to CANDOR. It is also not true, for example Otter [5] uses execution feedback and heterogenous prompts for test case generation on repository-level tasks.

**Lack of evaluation diversity (benchmarks, LLMs)**
- The method evaluates on only two different model families: GPT-4.1 (mini/nano) and DeepSeek-R1-Distill-Qwen (7B/32B), both reasoning trained LLMs. The evaluation lacks any other open-source and closed source models (Mistral, Llama, Claude, Gemini, to name a few from the top of my head), in particular SOTA models (e.g., R1, GPT-4.1) or any non-reasoning LLMs or LLMs trained specifically for coding and provides no justification to exclude them (e.g. they saturate the benchmark already)
- The method is only evaluated on single function benchmarks. This results in the method being much less relevant, as nowadays benchmarks on repository level [2,3] are already being adressed by more advanced systems.
- The execution feedback based on a candidate implementation is highly contingent on the capability of the model to generate a correct solution (especially as the solution can not be adapted anymore based on the oracle feedback! only the other way around). This strongly skews the results, especially as HumanEval and MBPP and similar benchmarks are known to be somewhat saturated by recent models [4], thus making it easier to use generated solutions as trustworthy execution feedback. Evaluating on other benchmarks where generating a correct candidate implementation is not as easily possible [2,3] would be able to resolve this concern.


**Further Nitpicks**

- The code submission is not documented. It should contain a clear documentation on how to set up an environment to execute the scripts (which python version, packages etc) and how to run the scripts to reproduce the results in the paper (including baselines and other works). In the current state, I would mark this research as "not reproducible"
- The authors could use \paragraph{...} to structure long sections, for example \paragraph{Related works lacks diversity} to start the paragraph in line 77, similarly line 86 and line 99 ("this work does ...", "evaluation shows ...")
- I could not find an explicit explanation of the difference between task-level and test-level scores (Table 2). Please explicitly describe what each metric measures in the evaluation metrics paragraph (e.g., a formula that unambiguously defines the metric)
- The writing could use additional passes to increase conciseness.


References
[1] Molina et al, Test Oracle Automation in the Era of LLMs, 2025
[2] Mündler et al, SWT-Bench: Testing and Validating Real-World Bug-Fixes with Code Agents, NeurIPS 2024
[3] Jain et al, TestGenEval: A Real World Unit Test Generation and Test Completion Benchmark, ICLR 2025
[4] Yu et al, HumanEval Pro and MBPP Pro: Evaluating Large Language Models on Self-invoking Code Generation, ACL 2025
[5] Ahmed et al, Generating Tests from Issues to Validate SWE Patches, ICML 2025

**Questions:**

Please see the questions raised in the weaknesses section.

---

> ### Author Response · Authors · 2025-11-19
>
> **W1.1: Table 6 only shows performance difference after a round of debugging information after direct generation/CANDOR/Nexus. Please also include the performance without debugging information as a true baseline so that it can be seen whether the oracles increase performance at all.**
>
> Response:
> We clarify that, as mentioned in Lines 408-410, the evaluation of bug detection (Table 6) and the debugging process (Table 7) are conducted on our constructed **buggy code set**, where each of the solutions is already incorrect by the verification of the dataset's provided private test cases. Therefore, the performance without debugging information would be **0%** for all datasets.
>
> **W1.2: The method only allows adapting the tentative oracles based on execution feedback of the candidate solution, but not to adapt the candidate solution based on the generated oracles.**
>
> Response:
> This is a good question. First, the key reason for us to use the LLM-generated code to adapt the tentative oracle, but not the code itself, is that this work **focuses on test oracle generation**. Specifically, similar to existing works on self-refine/debug-based code generation, which first generate candidate tests and then use these test cases to debug and refine the LLM-generated code, we transfer the optimization object from the code to the oracles. Next, we do not treat the LLM-generated code as fully correct (just as code generation works would not treat tests generated by LLMs as fully correct). We treat the LLM-generated code as having high confidence in its correctness, and then we use this code to further check the correctness of the generated oracle. Furthermore, as our experiments show (see the case study and Table 2), even when the provided code solution is incorrect, LLMs still have the ability to ensure that a correct test oracle (which may initially fail due to the incorrect code solution) is preserved after the self-refine stage. Finally, for the self-refine component in Nexus, as our results in Table 2 demonstrate that it can recover some incorrect oracles to correct ones, we believe that this component has value. To address the reviewer’s concern, we will provide an **argparse** setting in the final code to allow users to decide whether to use this self-refine component.
>
> Note: The *test_examples* in Appendix B.6 are prompted from the task description of the code generation works. For example, for HumanEval/Task1, two test examples, which are visible to the LLMs during the code generation process, already exist:
> - has_close_elements([1.0, 2.0, 3.0], 0.5)
> False
> - has_close_elements([1.0, 2.8, 3.0, 4.0, 5.0, 2.0], 0.3)
> True
>
> | **Scenario** | **Count / %** |
> |--|--|
> | A: **True Negatives** (Oracle was *Incorrect*, Candidate was *Correct*) | 6.5% |
> | B: Successful *Correction* (Refinement loop *fixes* the incorrect oracle) | 88.3% (of A) |
> | C: **False Negatives** (Oracle was *Correct*, Candidate was *Incorrect*) | 16.6% |
> | D: Successful *Recovery* (Refinement loop *preserves* the correct oracle) | 55.4% (of C) |
>
> *Table 2: Analysis of Oracle's Failing Validation Phase (HumanEval, GPT-4.1-Nano)*
>
> **W1.3: The authors do not provide the prompt for the direct generation baseline.**
>
> Response:
> Good question! **For Direct Generation, the sentence `Be quick, but it may not be perfect` does not exist in the prompt**, while other instructions are the same as the tentative prompt in Nexus.
> The sentence `Be quick, but it may not be perfect` is from Candor; we just follow its setup. The key reason for these prompts is to ensure that the reasoning LLMs do not consume excessive tokens during the tentative oracle generation process.
>
> **W1.4: The ablation for different agents does just remove critics from the nexus setups (sec 5.2). I think it would be cleaner to replace them with a generically prompted agent, strengthening the claim that the improvements in Nexus stem from more specific/diverse prompts. Currently, it mainly shows that fewer critics/fewer inputs to aggregate over reduce the performance of nexus.**
>
> Response:
> Good question! Actually, the key reason for us not to choose this process (i.e., introducing a new generically prompted agent) is that this agent would naturally become another agent in the planning process, meaning the number of agents we need to evaluate would change from 4 to 5. If we were to evaluate these 5 agents, the number of agents would further change from 5 to 6, leading to an unnecessary discussion spiral. Therefore, in our work, to avoid this, we just follow existing ablation practices by removing one of the planning agents and then analyzing the LLM's performance.

---

> > ### Author Response · Authors · 2025-11-19
> >
> > **W1.5: Why exactly were the 4 agents chosen that nexus employs? Do they result from the preliminary study, or are they chosen based on a validation setting, or based on educated guesses by the authors on what might be useful? Were other critics tried?**
> >
> > Response:
> > Good question. The main reason is that some of the contributors to this work have a background in the software engineering community. Based on their comments, we listed all commonly used rules to measure the test oracle and summarized these rules, finally forming the paper's four agents.
> >
> > **W2: Lack of comparison to related work and Lack of evaluation diversity (Baseline and benchmark)**
> > Response:
> > Thanks for your valuable suggestion! Following your suggestion, we conducted a new experiment on TestGenEval [1]. We chose TestGenEval as it is built from SWE-Bench (addressing the reviewer's request for more realistic benchmarks) and uses a pass@1 metric that aligns well with our evaluation. Given the high computational cost of running this benchmark, we evaluated all frameworks on a randomly selected subset of 164 tasks. As requested, we also included comparisons to strong, relevant baselines like SWT-Agent [2] and Otter [3]. The results in Table 4 demonstrate that Nexus outperforms all baselines, including those specifically designed for SWE-type tasks. This provides strong evidence that our framework's approach is not limited to academic benchmarks and generalizes effectively to more complex, real-world test generation scenarios.
> >
> > | Method | Pass@1 (%) |
> > |--|--|
> > | GPT-4.1-Nano | 50.6 |
> > | Candor | 62.8 |
> > | SWT-Agent | 54.3 |
> > | Otter | 65.2 |
> > | Nexus | 67.1 |
> >
> > *Table 4: Pass@1 (%) on TestGenEval.*
> >
> > **W3: Additional LLMs**
> >
> > Response:
> > Great question! Our choice of LLMs first involves analyzing whether they are reasoning LLMs, and then considering their price and common usage.
> > Since agent frameworks usually work for reasoning LLMs or LLMs with very large parameters (e.g., CoT requires LLMs > 175B parameters [4]), we did not choose LLMs that lack reasoning ability, such as Code Llama. Next, as we do not have enough funding for a large-scale evaluation of very expensive commercial SOTA LLMs (e.g., o-version GPT models), we chose GPT-4.1-Nano and GPT-4.1-Mini as our base LLMs because they are cost-effective and more commonly used by developers.
> >
> > To address the reviewer’s concern about the generalizability of Nexus on other LLMs, we provide evaluation results below on TestEval and LiveCodeBench (two hard test generation benchmarks) with two additional LLMs.
> >
> > | | TestEval | LiveCodeBench |
> > |--|--|--|
> > | GPT-4.1 | 66.8 | 61.3 |
> > | +Nexus | 70.7 (+3.9) | 74.2 (+12.9) |
> > | GPT-4-Sonnet | 56.7 | 43.4 |
> > | +Nexus | 82.1 (+25.4) | 56.6 (+13.1) |
> >
> > **Nitpicks:**
> > Thanks for your valuable suggestions. For the **task-level accuracy**, we analyze whether *all* test oracles for the task (e.g., 20 in our manuscript) are correct. If *any* oracle is incorrect, we treat the LLM-generated test oracles for that task as incorrect.
> > For the **test-level accuracy**, we analyze each test oracle's correctness individually. For example, if 10 out of 20 oracles for the task are correct, then the test-level accuracy is 50% (10 out of 20).
> >
> > [1] Jain, Kush, Gabriel Synnaeve, and Baptiste Roziere. "Testgeneval: A real world unit test generation and test completion benchmark." arXiv preprint arXiv:2410.00752 (2024).
> >
> > [2] Mündler N, Müller M, He J, et al. SWT-bench: Testing and validating real-world bug-fixes with code agents[J]. Advances in Neural Information Processing Systems, 2024, 37: 81857-81887.
> >
> > [3] Ahmed, T., Ganhotra, J., Pan, R., Shinnar, A., Sinha, S., & Hirzel, M. (2025). Otter: Generating Tests from Issues to Validate SWE Patches. arXiv preprint arXiv:2502.05368.
> >
> > [4] Wei, J., Wang, X., Schuurmans, D., Bosma, M., Xia, F., Chi, E., ... & Zhou, D. (2022). Chain-of-thought prompting elicits reasoning in large language models. Advances in neural information processing systems, 35, 24824-24837.

---

> ### Comment · Reviewer_6rvk · 2025-11-26
>
> I generally thank the authors for their extensive answer. I will list below the remaining open questions after reading through the answer.
>
> **W1.1, W1.2**: I thank the authors for clarifying that the presented data is only on buggy programs and I apologize for the confusion. In this case the presented numbers in Table 6/7 make sense. However, I was missing data on how likely it would be for the model to break correct oracles based on an incorrect candidate solution. As shown in the answer to W1.2, this appears quite common (45% of incorrect candidate solutions break the oracle). Therefore, if the model struggles to come up with a good candidate solution (lets say the base rate of incorrect candidate solutions would be closer to 80%, such as for example on the SWE-Bench tasks), Nexus would likely break many of the (possibly correctly) generated oracles. Recovering from incorrect candidates would hence be a very desirable property.
>
> What would mitigate my concerns is a clarification about what is the overall base rate of correct candidates on the benchmarks (including TestGenEval)? What is the performance on Nexus when the base rate of correct candidates is low?
>
> **W1.4, W1.5**:
> I think it would be a good ablation to try different agents. First, it is not necessarily true that a combination of all possibible agents is actually best (i.e., I disagree with your answer on W1.4) and second, this would indicate that your method is much stronger than the specific data point provided: It would be a general framework into which agents can be inserted/removed/swapped to further tune performance.
>
> **W2**:
> These results are indeed promising. However, why did you not evaluate on SWT-Bench, which is the more realistic variant IMO, and what Otter and SWT-Agent where designed for? Is your method not applicable to this setting?
>
> **Presentation**:
> Finally, I pointed out a number of issues with the presentation and writing. Are the authors able to upload a revised PDF that outlines their intended changes or at least confirm which changes they plan for the camera ready revision?
>
> ----
> Overall, given the new results, I have increased my score. I think the paper is below the standards for the conference especially regarding generality, novelty and presentation. However, they *do* appear to get better results than prior work, which I understand appeals to certain audiences.

---

> > ### Author Response · Authors · 2025-11-27
> >
> > **W1.1, W1.2: What would mitigate my concerns is a clarification about what is the overall base rate of correct candidates on the benchmarks (including TestGenEval)? What is the performance on Nexus when the base rate of correct candidates is low?**
> >
> > We thank the reviewer for this insightful follow-up. We agree that recovering from incorrect candidates is a critical property. To address your concern about the base rate of correct candidates, we analyzed the raw Pass@1 accuracy of the candidate solutions (generated by the base LLMs) across all benchmarks.
> >
> > First, as you hypothesized, the base rate of correct candidates drops significantly on more complex benchmarks. As shown in the table below, on TestGenEval, the candidate accuracy is only 19.51\% (GPT-4.1-Nano) and 23.70\% (GPT-4.1-Mini) for our evaluation subset. Similarly, on LiveCodeBench, the accuracy is as low as 12.15%. This confirms your intuition that these environments provide a rigorous test bed where incorrect candidates are the norm.
> >
> > | Model | HE | HE-Pro | MBPP | MBPP-Pro | TE | LCB | ULT | **TestGenEval** |
> > | :--- | :--- | :--- | :--- | :--- | :--- | :--- | :--- | :--- |
> > | **GPT-4.1-Nano (Candidate)** | 73.78 | 65.85 | 78.60 | 65.61 | 83.41 | **12.15** | 82.05 | **19.51** |
> > | **GPT-4.1-Mini (Candidate)** | 92.68 | 73.78 | 87.55 | 75.93 | 81.95 | **17.76** | 91.45 | **23.70** |
> >
> > However, despite the base candidate accuracy being extremely low on TestGenEval and LCB, Nexus achieves a Pass@1 of 67.1% and 56.71%. This result directly mitigates the concern that Nexus simply "breaks" correct oracles to fit incorrect candidates. If Nexus were heavily dependent on the correctness of the initial candidate, we would expect its performance to degrade toward the base rate. Instead, we observe a massive performance gain (+11.14%  on LCB for GPT-4.1-Mini). This demonstrates that Nexus possesses the robust "recovery" property described before: it effectively utilizes its verification and refinement loops to overcome the high prevalence of incorrect candidate solutions, rather than being misled by them (as we also requires the LLMs analyze whether the error information are correct, if it believe it is not correct, the LLMs can still return the original oracle).
> >
> >
> > **W1.4, W1.5: I think it would be a good ablation to try different agents. First, it is not necessarily true that a combination of all possibible agents is actually best (i.e., I disagree with your answer on W1.4) and second, this would indicate that your method is much stronger than the specific data point provided: It would be a general framework into which agents can be inserted/removed/swapped to further tune performance.**
> >
> > Response:
> > We appreciate the reviewer’s insightful suggestion to treat NEXUS as a general framework and test its robustness by modifying the agent composition. We agree that the value of our method lies in its modularity rather than just a fixed configuration. To verify this, we conducted additional ablation studies on the HumanEval and HumanEval-Pro benchmarks. We tested two scenarios for every agent (SE, ES, FV, AA): (1) removing the agent entirely, and (2) replacing the specialized agent with a generic "common panelist (prompt: Generate test assertions for the following function)."
> >
> > The results support the reviewer's hypothesis that NEXUS functions as a flexible general framework. As shown in the table, the system remains effective even when agents are swapped. For instance, replacing the Adversarial Attacker (AA) with a common panelist recovers significant performance (92.74% on HumanEval) compared to simply removing it (89.36%). However, the full NEXUS configuration (combining all specialized agents) still achieves the highest performance (93.69% on HumanEval and 79.47% on HumanEval-Pro). This demonstrates that while the framework is general enough to accommodate different agent configurations, the specific specialized roles we designed provide a distinct performance advantage over generic agents.
> >
> > | Setting | HE (Remove) | HE (Replace) | HE-Pro (Remove) | HE-Pro (Replace) |
> > | :--- | :---: | :---: | :---: | :---: |
> > | **NEXUS (Full)** | **93.69** | - | **79.47** | - |
> > | w/o SE | 91.62 | 92.47 | 73.47 | 78.36 |
> > | w/o ES | 91.22 | 92.41 | 74.18 | 78.33 |
> > | w/o FV | 90.88 | 92.29 | 74.24 | 78.36 |
> > | w/o AA | 89.36 | 92.74 | 72.45 | 77.80 |

---

> > > ### Author Response · Authors · 2025-11-27
> > >
> > > **W2: These results are indeed promising. However, why did you not evaluate on SWT-Bench, which is the more realistic variant IMO, and what Otter and SWT-Agent were designed for? Is your method not applicable to this setting?**
> > >
> > > Response:
> > > We appreciate the reviewer suggesting SWT-Bench. To answer your first question: Yes, our method is fully applicable to the SWT-Bench setting. We agree that SWT-Bench is a realistic variant; however, we utilized TestGenEval for this evaluation because it is also directly derived from SWE-Bench and serves as an equally rigorous proxy for repository-level test generation. **Both SWT-Bench and TestGenEval share the same lineage and complexity from SWE-Bench**, effectively measuring the same underlying capabilities for test case generation.
> > >
> > > Regarding the choice of dataset, we prioritized TestGenEval during the rebuttal phase due to execution stability. While attempting to deploy SWT-Bench, we encountered significant compatibility issues with several of its provided Docker containers on our infrastructure. TestGenEval, conversely, provides Docker images where dependencies and requirements have been manually pre-fixed and stabilized (*Modify SWEBench images: We start with the Docker images provided by SWEBench. We first modify the images that did not build manually, by installing appropriate dependencies and modifying requirements files so every test suite executes*)). To ensure we could provide the reviewer with reliable, complete experimental results within the rebuttal timeframe, we proceeded with the more stable TestGenEval environment. We are currently resolving the environment configuration issues for SWT-Bench and commit to including these results in the camera-ready version to further demonstrate the generalizability of Nexus.
> > >
> > > **Presentation: Finally, I pointed out a number of issues with the presentation and writing. Are the authors able to upload a revised PDF that outlines their intended changes or at least confirm which changes they plan for the camera ready revision?**
> > >
> > > Response:
> > > We appreciate your constructive feedback on the manuscript's presentation. In response to your request, we have uploaded a revised version of the paper. This updated PDF addresses the writing errors and presentation issues you pointed out, ensuring the final camera-ready version will meet the required standards.
> > >
> > > ---
> > >
> > > We sincerely thank the reviewer for re-evaluating our work and increasing the score. We believe that the new ablation studies (demonstrating Nexus as a flexible, general framework) and the uploaded revised manuscript (which resolves the presentation issues and clarifies the novelty of our error-recovery mechanism) have significantly strengthened the paper in the areas you identified.

---

> ### Comment · Reviewer_6rvk · 2025-11-27
>
> I thank the authors for the swift and detailed response. The revised version is significantly improved over the initial submission. While I disagree that the authors provided significant evidence for novelty (in particular they did not prove that their contributions enabled error recovery as mentioned, this would require a comparison to the error recovery capabilities of the previously mentioned CANDOR), I think the new results significantly strengthen the provided method. Therefore, I am quite confident now that the provided numbers indeed indicate a definite improvement over prior methods.
>
>
> That being said, my concern for novelty remains, although I regret not formulating it out in the initial review. Fundamentally, both CANDOR (which is not published) and NEXUS are just applications of Multi-Agent Systems and Execution Feedback to the problem of Test Generation.
>
> Multi Agent Systems have been around for a while [1], Execution Feedback as well [2]. Neither is the presented work the first to attempt solving Test Generation with LLMs/Agents/Execution Feedback, with all of Otter (2025), SWT-agent (2024) and CANDOR (arguably concurrent, but apparently an important basis for this work) existing.
>
> I therefore still lean on the side of rejection but would not mind acceptance if other reviewers have strongly different opinions (there appear to be opinions in either direction).
>
> [1] Gui et al, *Large Language Model based Multi-Agents: A Survey of Progress and Challenges*, arXiv
> [2] Liu et al, *A Survey on the Feedback Mechanism of LLM-based AI Agents*, arXiv

---

> ### Author Response · Authors · 2025-11-28
>
> We thank the reviewer for increasing the overall score to 4 and for acknowledging that the revised version is significantly improved over the initial submission.
>
> **Comment: They did not prove that their contributions enabled error recovery as mentioned; this would require a comparison to the error recovery capabilities of the previously mentioned CANDOR**
>
> Response:
> Regarding the concern on novelty and the comparison with CANDOR: While we agree that both frameworks operate within the high-level paradigm of Multi-Agent Systems (MAS) and Execution Feedback, the architectural instantiation of these concepts is where Nexus contributes novelty. Fundamentally, the frameworks differ in their structural composition:
>
> - CANDOR: Tentative Oracles + Planning (3 generic panelists).
> - Nexus: Tentative Oracles + Planning (4 SE-specific panelists) + Refinement.
>
> Nexus advances the state-of-the-art by explicitly specializing the planning stage and introducing a decoupled error recovery phase. To demonstrate the isolated value of these contributions and answer the reviewer's request regarding error recovery, we provide the following component analysis (ablation study) on HumanEval and HumanEval-Pro.
>
> *Table: Component Analysis and Error Recovery Comparison*
>
> | Method Configuration | HumanEval | HumanEval-Pro |
> |----------------------|:---------:|:-------------:|
> | 1. Direct Generation (Baseline) | 27.44 | 10.37 |
> | 2. CANDOR (Existing SOTA) | 56.10 | 28.66 |
> | 3. Nexus (Planning Only) | 57.32 | 32.32 |
> | 4. CANDOR + Nexus Refinement | 59.76 | 34.76 |
> | 5. Nexus (Full: Planning + Refinement) | **65.85** | **41.46** |
>
> - Novelty of Planning (Row 2 vs. Row 3): Even without the refinement stage, Nexus’s planning module (utilizing four specialized agents derived from software engineering principles) outperforms CANDOR’s planning module (three generic panelists). On HumanEval-Pro, Nexus (Planning Only) achieves 32.32% compared to CANDOR's 28.66%, proving that our SE-driven agent design provides a stronger foundation than generic agent roles.
>
> - Novelty of Error Recovery (Row 2 vs. Row 4): We applied our Refinement (error recovery) module on top of CANDOR. This improved CANDOR’s performance from 28.66% to 34.76% on HumanEval-Pro. This demonstrates that our error recovery mechanism is a distinct, transferable contribution that effectively fixes errors where prior methods fail.
>
> - Combined Synergy (Row 5): The full Nexus framework achieves the highest accuracy (41.46% on HumanEval-Pro), confirming that the combination of specialized SE-planning and iterative refinement offers significant advantages over existing concurrent works.
>
> We hope this breakdown clarifies that Nexus is not merely a generic application of MAS, but a specific, novel architecture that outperforms the concurrent CANDOR framework in both planning efficacy and error recovery.

---

### Official Review · Reviewer_Rmz8 · 2025-10-31

**Soundness:** 3
**Presentation:** 4
**Contribution:** 3
**Rating:** 8
**Confidence:** 4

**Summary:**

This paper presents Nexus, a novel multi-agent framework addressing the difficult challenge of test oracle generation: determining whether a function’s output matches its intended behavior. Nexus generates test oracles by leveraging multi-agent collaboration through a cycle of deliberation, validation, and self-refinement.
Nexus introduces several innovations:
- A multi-agent debate among specialized testing personas (Specification Expert, Edge Case Specialist, Functional Validator, Algorithmic Analyst), each offering a distinct point of view when assessing the quality of a candidate oracle.
- An execution-grounded feedback mechanism, where candidate oracles are validated through actual test execution within a sandbox.
- An iterative self-refinement process, in which failed oracles are automatically improved using LLM feedback.

The framework achieves consistent improvements across seven benchmarks on oracle accuracy; additionally, it helps improving on downstream tasks such as bug detection and automated program repair.

**Strengths:**

1. The paper is well written and easy to follow, with a good balance between qualitative examples and quantitative evidence. The problem is well motivated as well: Table 1 convincingly illustrates that while LLMs can already generate syntactically valid test inputs, they still struggle with semantic reasoning to produce correct outputs.
2. The experimental evaluation is comprehensive, involving multiple open and closed source LLMs. Results across several benchmarks are consistent and demonstrate substantial improvements.
3. This work shows not only improvement on test oracle generation, but also highlights how downstream tasks, such as bug detection and automated program repair, can benefit from this.
4. Novelty on top of prior work (e.g., CANDOR):
    - expanded deliberation phase with different testing personas
    - execution-based validation and iterative self-refinement, grounding reasoning in executable evidence.
5. Each component (deliberation, validation, self-refinement) is thoroughly analyzed, showing measurable contributions to the overall performance.

**Weaknesses:**

1. Nexus is more complicated than prior frameworks. While the performance improvements are clear, the paper lacks a detailed analysis of computational cost (e.g., API budget, runtime).
2. The validation phase (lines 213–216) relies on an LLM-generated "candidate implementation" of the function under test. If this implementation is incorrect, it could lead to false positives or negatives: more analysis is needed to better understand how the oracle would behave in such a case.
3. The comparison focuses primarily on CANDOR, but Nexus is not evaluated on all benchmarks used by that baseline (e.g., HumanEvalJava, LeetCodeJava).
4. Incorporating additional baselines, such as those appearing in SWT-Bench or other relevant test-generation benchmarks, would improve the completeness and fairness of the experimental section.

**Questions:**

- Lines 216-217 mention that passing oracles are considered correct: more analysis on this would help understand how much this hypothesis is correct.
- It would be interesting to see more qualitative examples from the preliminary analysis in the appendix.
- Question related to weakness #3: are baselines missing for HumanEvalJava and LeetCodeJava because Nexus has been tested only on Python? What would be the generalizability of the framework?
- Question related to weakness #4: How is CANDOR established as the state-of-the-art baseline? It does not appear in public leaderboards such as LiveCodeBench or SWT-Bench. The paper might benefit from a comparison with other baselines.
- Some of the datasets are more academic (e.g., HumanEval, MBPP): in the past, models were achieving high scores on such benchmarks but very low ones when tested on more realistic ones (e.g., SWE-Bench). Have the authors considered evaluating Nexus on more real-world benchmarks (e.g., SWT-Bench, SWE-Bench)?
- Could the paper include qualitative success and failure cases of Nexus to illustrate its reasoning and limitations better?

---

> ### Author Response · Authors · 2025-11-19
>
> We thank Reviewer Rmz8 for the positive feedback on our performance improvements and for the constructive suggestions.
>
> **W1: The paper lacks a detailed analysis of computational cost (e.g., API budget, runtime).**
>
> This is a fair and important point. Nexus intentionally trades computation for higher accuracy. To quantify this trade-off, we provide the detailed cost analysis for the HumanEval evaluation using GPT-4.1-Nano for each framework with the average token usage, runtime, and estimated API cost below:
>
> | **Framework** | **Avg. Tokens (M)** | **Avg. Runtime (s)** | **Est. API Cost ($0.15/1M in, $0.60/1M out)** |
> | :--- | :---: | :---: | :---: |
> | Direct Generation | 0.2 | 15.42 | $0.06 |
> | CANDOR | 3.5 | 67.28 | $1.06 |
> | Nexus (Ours) | 4.8 | 87.63 | $1.42 |
>
> *Table 1: Computational cost of Nexus and baselines.*
>
> **W2&Q1: If this implementation is incorrect, it could lead to false positives or negatives: more analysis is needed to better understand how the oracle would behave in such a case.**
>
> This is a good and critical question shared by several reviewers, and we will clarify our position in Section 3.3. We do not treat the LLM-generated candidate implementation as a perfect ground truth. **Its role is to serve as a plausible validator, not a perfect oracle;** this is conceptually similar to code generation works that use an LLM-generated test case to debug generated code. The goal of this phase is **falsification:** the candidate provides an execution-based check that is orthogonal to the deliberation-based check from Phase 1. An oracle that passes both abstract reasoning and this concrete execution check is highly likely to be correct. The primary benefit is catching incorrect oracles that passed deliberation (a **True Negative**), allowing the refinement loop to fix them. However, we also acknowledge the reviewer's specific concern about **False Negatives** (i.e., what happens if an incorrect candidate falsely fails a correct oracle?). This scenario is precisely why we designed the iterative self-refinement loop (Phase 3). The refinement agent (Prompt B.7) is given the candidate code, the failed oracle, and the specific error, allowing it to reason that the candidate may have misunderstood the specification and that the original oracle was, in fact, correct. Our existing evidence in Table 3 already shows that this Refinement phase provides a significant complementary gain over Planning (deliberation) alone.
>
> To provide a more direct analysis of both scenarios, we also conduct a new detailed case study on HumanEval (using GPT-4.1-Nano). As shown in the new experiment table below, this analysis quantifies both the intended benefit (fixing incorrect oracles) and the robustness to False Negatives, providing strong quantitative evidence that the pipeline works as intended and is robust to the exact failure mode the reviewer described.
>
> | **Scenario** | **Count / %** |
> | :--- | :---: |
> | A: **True Negatives** (Oracle was *Incorrect*, Candidate was *Correct*) | 6.5% |
> | B: Successful *Correction* (Refinement loop *fixes* the incorrect oracle) | 88.3% (of A) |
> | C: **False Negatives** (Oracle was *Correct*, Candidate was *Incorrect*) | 16.6% |
> | D: Successful *Recovery* (Refinement loop *preserves* the correct oracle) | 55.4% (of C) |
>
> *Table 2: Analysis of Oracle's Failing Validation Phase (HumanEval, GPT-4.1-Nano)*
>
> **W3&Q3: Are baselines missing for HumanEvalJava and LeetCodeJava because Nexus has been tested only on Python? What would be the generalizability of the framework?**
>
> Response: In our manuscript, we demonstrate Nexus's performance by evaluating it on multiple diverse test generation benchmarks like MBPP, TestEval, and UTL.
> To address the reviewer’s concern about Nexus’s performance on Java tasks, we follow CANDOR’s setup and select 100 tasks from LeetCode Java solutions (as CANDOR does not release its Java tasks, we then randomly select 100 tasks from Venus [1]). Using the same experimental setup (with GPT-4.1-Nano) as our main experiments, the results below show that Nexus outperforms both baselines, supporting the generalizability of our conceptual framework.
>
> | Method | Test-Level Accuracy (%) |
> |--|--|
> | GPT-4.1-Nano | 42.2 |
> | Candor | 48.7 |
> | Nexus | 51.3 |
>
>
>
> *Table 3: Test-Level Accuracy on LeetCode Java with GPT-4.1-Nano*

---

> > ### Author Response · Authors · 2025-11-19
> >
> > **W4&Q4&Q5: Additional Baselines and Real-World Benchmarks.**
> > Response:
> > Thank you for this valuable suggestion. We chose CANDOR as our primary SOTA baseline because it is the most recent and direct-comparison work for multi-agent LLM-based test oracle generation. We agree that comparing Nexus against baselines on real-world benchmarks is crucial. Following your suggestion, we conducted a new experiment on TestGenEval [2]. We chose TestGenEval as it is built from SWE-Bench (addressing the reviewer's request for more realistic benchmarks) and uses a pass@1 metric that aligns well with our evaluation. Given the high computational cost of running this benchmark, we evaluated all frameworks on a randomly selected subset of 164 tasks. As requested, we also included comparisons to strong, relevant baselines like SWT-Agent [3] and Otter [4]. The results in Table 4 demonstrate that Nexus outperforms all baselines, including those specifically designed for SWE-type tasks. This provides strong evidence that our framework's approach is not limited to academic benchmarks and generalizes effectively to more complex, real-world test generation scenarios.
> >
> > | Method | Pass@1 (%) |
> > |--|--|
> > | GPT-4.1-Nano | 50.6 |
> > | Candor | 62.8 |
> > | SWT-Agent | 54.3 |
> > | Otter | 65.2 |
> > | Nexus | 67.1 |
> >
> > *Table 4: Pass@1 (%) on TestGenEval.*

---

> > > ### Author Response · Authors · 2025-11-19
> > >
> > > **Q2&Q6: On Qualitative Examples and Success/Failure Cases.**
> > >
> > > Thanks for your valuable suggestion! To illustrate Nexus's reasoning and how it handles the False Negative scenario (W2&Q1), we will add the following detailed qualitative case study to the Appendix.
> > >
> > > To illustrate the full Nexus pipeline, we analyze the task findAnswer(n, edges), which requires identifying all edges that are part of any shortest path from node 0 to n-1. We will examine two key scenarios using two different LLM-generated candidates: **Correct Candidate** (e.g., GPT-4.1-Mini): A correct implementation that runs Dijkstra from both node 0 and node n-1. **Incorrect Candidate** (e.g., GPT-4.1-Nano): A flawed implementation that only runs Dijkstra from node 0.
> > >
> > > Scenario 1: True Negative (Fixing a Buggy Oracle). This case demonstrates Nexus's primary goal: using a correct candidate to find and fix a flawed oracle.
> > >
> > > Task: findAnswer(3, [[0,1,1], [1,2,1], [0,2,3]])
> > >
> > > Correct Output: [True, True, False] (The shortest path is 0->1->2, cost 2. The direct edge 0->2 has cost 3 and is not a shortest path).
> > >
> > > Phase 1: Deliberation
> > >
> > > Tentative Oracle: Let's assume the initial prompt (or a flawed panelist) generates an incorrect oracle:
> > > assert s.findAnswer(...) == [True, True, True]
> > >
> > > Flawed Reasoning: "The path 0->1->2 (cost 2) is a shortest path, so edges [0,1,1] and [1,2,1] are True. The path 0->2 (cost 3) is also a path, so [0,2,3] is also True." This logic fails to check if the 0->2 path is the shortest one.
> > >
> > > Phase 2: Validation
> > >
> > > Candidate: The **Correct Candidate** (GPT-4.1-Mini) is used.
> > >
> > > Execution: The correct candidate runs and correctly determines the output is [True, True, False].
> > >
> > > Result: AssertionError: [True, True, True] != [True, True, False]. The oracle FAILS.
> > >
> > > Phase 3: Self-Refinement
> > >
> > > Input to LLM: "Fix this failed oracle. Candidate code is [Correct Code]. Failed oracle ...==[True, True, True] produced error ...!= [True, True, False]."
> > >
> > > LLM Reasoning: "The error shows the candidate returned [False] for the third edge [0,2,3]. The candidate code correctly finds the shortest path cost is 2 (via 0->1->2). The edge [0,2,3] has a cost of 3, which is greater than 2. Therefore, it is not part of any shortest path. The candidate's output [True, True, False] is correct, and the original oracle [True, True, True] was wrong."
> > >
> > > Final Oracle: assert s.findAnswer(...) == [True, True, False] (Corrected)
> > >
> > > Scenario 2: False Negative (Recovering from a Buggy Candidate)
> > >
> > > This case directly addresses the reviewers' concern (W2&Q1) and demonstrates Nexus's robustness.
> > >
> > > Task: findAnswer(3, [[0,1,2], [1,2,2], [0,2,10]])
> > >
> > > Correct Output: [True, True, False] (The shortest path is 0->1->2, cost 4. The edge [0,2,10] is not a shortest path).
> > >
> > > Phase 1: Deliberation
> > >
> > > Tentative Oracle: The panel correctly identifies the shortest path 0->1->2 and generates the correct oracle:
> > > assert s.findAnswer(...) == [True, True, False]
> > >
> > > Phase 2: Validation
> > >
> > > Candidate: The **Incorrect Candidate** (GPT-4.1-Nano) is used (only runs dijkstra_from_zero).
> > >
> > > Execution: The flawed candidate calculates dist0 = [0, 2, 4]. It then incorrectly checks edge [1,2,2] using dist0[1] + 2 + dist0[2] == 4, which evaluates to 2 + 2 + 4 = 8. This check fails, and the candidate incorrectly returns [True, False, False].
> > >
> > > Result: AssertionError: [True, True, False] != [True, False, False]. The correct oracle FAILS due to the incorrect candidate.
> > >
> > > Phase 3: Self-Refinement
> > >
> > > Input to LLM: "Fix this failed oracle. Candidate code is [Incorrect "Nano" Code]. Failed oracle ...==[True, True, False] produced error ...!= [True, False, False]."
> > >
> > > LLM Reasoning: "The error shows the candidate returned [False] for the second edge [1,2,2]. Analyzing the candidate's code, it only runs Dijkstra from node 0. Its logic for checking edge (a, b, w) seems to be dist0[a] + w + dist0[b] == target. This is algorithmically incorrect for finding all shortest path edges. The correct method requires running Dijkstra from both 0 and n-1. The original oracle's claim that [True, True, False] is the answer implies edge [1,2,2] is part of the shortest path (0->1->2), which is correct. The candidate's logic is flawed."
> > >
> > > Final Oracle: assert s.findAnswer(...) == [True, True, False] (Preserved)
> > >
> > > ---
> > >
> > > [1] Du, Mingzhe, et al. "Mercury: A code efficiency benchmark for code large language models." Advances in Neural Information Processing Systems 37 (2024): 16601-16622.
> > >
> > > [2] Jain, Kush, Gabriel Synnaeve, and Baptiste Roziere. "Testgeneval: A real world unit test generation and test completion benchmark." arXiv preprint arXiv:2410.00752 (2024).
> > >
> > > [3] Mündler N, Müller M, He J, et al. SWT-bench: Testing and validating real-world bug-fixes with code agents[J]. Advances in Neural Information Processing Systems, 2024, 37: 81857-81887.
> > >
> > > [4] Ahmed, T., Ganhotra, J., Pan, R., Shinnar, A., Sinha, S., & Hirzel, M. (2025). Otter: Generating Tests from Issues to Validate SWE Patches. arXiv preprint arXiv:2502.05368.

---

### Meta-Review · Area_Chair_yfvz · 2026-01-05

**Summary:**

1. Bias due to using an LLM-generated "candidate implementation" (most important).
2. Concerns about computational cost.
3. More baselines would be better.
4. Insufficient comparison to related work.
5. Limited practical applicability.

**Reviewer Concerns:**

1. Bias due to using an LLM-generated "candidate implementation" (most important). -> addressed empirically to an extent, not addressed theoretically
2. Concerns about computational cost. -> addressed
3. More baselines would be better. -> addressed
4. Insufficient comparison to related work. ->addressed
5. Limited practical applicability. -> not addressed

**Reviewer Scores:**

Rmz8: still 8
6rvk: 2->4 (because some additional evidence was provided)
zCg8 still 4 (concerns about evaluation remain) Note: the authors claim the reviewer raised their score. I can't see if they did and in any case Iam asked my own opinion.
bF44: still 2 (main concern about real-world applicability unresolved)

---

### Decision · Program_Chairs · 2026-01-26

Reject